# PatternKV: Flattening KV Representation Expands Quantization Headroom

## Abstract

KV cache in autoregressive LLMs eliminates redundant recomputation but has emerged as the dominant memory and bandwidth bottleneck during inference, notably with long contexts and test-time scaling. KV quantization is a key lever for reducing cache cost, but accuracy drops sharply as the native KV distribution lacks flatness and thus maintains a wide quantization range. Prior work focuses on isolating outliers, which caps their error but fails to flatten the overall distribution, leaving performance fragile under low-bit settings. In this work, we show that the K cache maintains a stable structure that evolves gradually with context, while the V cache carries latent semantic regularities. Building on these insights, we propose **PatternKV**, a pattern-aligned residual quantization scheme. It mines representative pattern vectors online, aligns each KV vector to its nearest pattern, and quantizes only the residual. This reshaping of the KV distribution flattens the quantization target and narrows its range, thereby improving the fidelity of low-bit KV quantization. Across long-context and test-time scaling settings on multiple backbones, PatternKV delivers consistent 2-bit gains, with a 0.08% average 4-bit drop relative to FP16, improves test-time scaling accuracy by 10% on average, and raises throughput by 1.4× while supporting 1.25× larger batches.

## 1 Introduction

Large language models (LLMs) have achieved remarkable performance in various tasks (OpenAI, 2023; Yang et al., 2024; Dubey et al., 2024; Jiang et al., 2023), yet such performance is grounded in autoregressive decoding. This process relies on a key-value (KV) cache to avoid redundant recomputation, but the cache itself has become a dominant memory and bandwidth bottleneck during inference (Kwon et al., 2023; Sheng et al., 2023). This challenge is further compounded by two key drivers: (i) **long contexts**, prevalent in tasks such as retrieval-augmented generation (Lewis et al., 2020) and long-document processing (Beltagy et al., 2020); and (ii) **test-time scaling**, arising from both long chain-of-thought reasoning (depth-oriented expansion) (Muennighoff et al., 2025), and multi-sample inference like self-consistency (Wang et al., 2023) or tree search (Xie et al., 2023; Wu et al., 2025) (breadth-oriented expansion). Taken together, these trends highlight the need for efficient yet high-fidelity KV cache compression in practical LLM deployment.

Quantization (Ashkboos et al., 2024; Frantar et al., 2022) is a widely adopted approach for KV cache compression, reducing memory footprint via lower-bit KV representations. The effectiveness of KV quantization largely depends on the flatness of the vector distribution: flatter distributions yield a narrower quantization range and preserve higher precision under limited bit widths. In pursuit of this, Hooper et al. (2024); Kang et al. (2024); Su et al. (2025) handle outliers by storing them with original precision, separated from the main KV distribution to minimize their impact on quantization. Meanwhile, Liu et al. (2024b) confines outlier-induced quantization error by quantizing key cache per-channel, ensuring the error remains within individual channels. However, these methods are primarily limited to protecting outliers rather than flattening the entire distribution.

In contrast, we tackle the root cause of quantization inefficiency by reshaping the entire KV distribution. Guided by a variance–decomposition perspective, we mine common patterns in KV caches, align each vector to its nearest pattern, and quantize only the residuals. This distribution-wide treatment flattens the quantization target, yielding narrower ranges and substantially reducing error under low-bit settings.

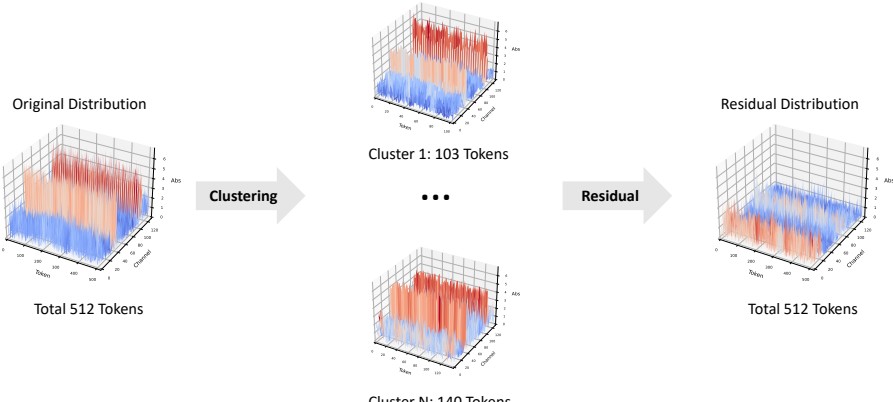

Figure 1: The left figure illustrates the original distribution of the KV vectors, while the right figure depicts the distribution of the residuals obtained after aligning the original vectors with the corresponding pattern vectors. Each pattern vector is the centroid of its cluster.

Specifically, our analysis of KV caches reveals exploitable regularities: the K cache maintains a stable structure but will evolve gradually with context, while the V cache exhibits latent semantic patterns. These findings indicate that pattern information can be reliably mined online without calibration corpora or additional tuning. Building on this, we employ clustering to extract representative pattern vectors that capture such common structure. During inference, each KV vector is aligned to its nearest pattern vector and transformed into a residual for quantization, resulting in a markedly flatter distribution. To accommodate the gradual evolution of KV distributions over decoding, we further introduce new pattern vectors on the fly, adaptively tracking shifts and maintaining quantization fidelity.

In summary, our main contributions are as follows:

- We introduce a variance–decomposition perspective on KV quantization, which shifts the focus from protecting outliers to flattening the overall distribution.

- We analyze latent patterns in the K and V caches, revealing stable structural and semantic regularities that motivate pattern-based residualization.

- We propose **PatternKV**, a lightweight, plug-and-play KV quantization scheme that improves low-bit accuracy with minimal overhead.

- We evaluate our method against strong baselines across diverse tasks and backbone models. In the long-context setting, our approach achieves consistent gains at 2-bit while limiting the 4-bit average drop relative to FP16 to just **0.08%**. Under test-time scaling, our method achieves a **10%** average improvement. In addition, our method achieves a **1.4×** throughput increase and supports a **1.25×** larger batch size.

## 2 MOTIVATIONS

### 2.1 A VARIANCE DECOMPOSITION VIEW OF KV QUANTIZATION

In KV cache quantization, asymmetric $n$-bit quantization is typically applied, with each vector $X$ mapped as:

$$Q(X) = \left\lfloor \frac{X - z}{s} \right\rceil, \qquad X_{\mathrm{deq}} = s \cdot Q(X) + z, \tag{1}$$

where $s = \frac{\max(X) - \min(X)}{2^n - 1}$ is the scaling factor and $z = \min(X)$ the zero-point, and $\lfloor \cdot \rceil$ denotes rounding to the nearest integer. The scaling factor $s$ critically determines quantization fidelity: a larger $s$ forces more distinct values into the same quantization level, while a smaller $s$ retains finer

distinctions. Therefore, flatter KV distributions with smaller ranges $max(X) - min(X)$ yield less distortion under quantization, and we use variance as a natural proxy for this flatness. This leads to the central question: **how can we reduce the variance of the K and V distributions to improve their quantization fidelity?**

The law of total variance (Blitzstein & Hwang, 2019) is widely used for analyzing variance reductions (Depeweg et al., 2018; Lakshminarayanan et al., 2017). It states that, given a partition of the data into groups, the total variance can be decomposed into two components: an intra-group term and an inter-group term. To apply this principle in the KV setting, we can introduce a set of representative pattern vectors $M$ that partition the collection of KV vectors into different clusters. Under this view, the total variance of KV vectors $Z$ decomposes as

$$\text{Var}(Z) = \underbrace{\mathbb{E}[\text{Var}(Z \mid M)]}_{\text{intra-pattern variance}} + \underbrace{\text{Var}(\mathbb{E}[Z \mid M])}_{\text{inter-pattern variance}} \tag{2}$$

The second term measures variance across pattern means. If we fix the pattern set $M$, the inter-pattern term vanishes. So the variance to be quantized reduces to $\mathbb{E}[\text{Var}(Z \mid M)]$. Therefore, the key to achieving a flatter quantization target lies in choosing a suitable partition that minimizes intra-pattern variance. In other words, the central challenge shifts from **reducing error on the raw distribution to selecting pattern vectors $M$ that yield a flatter quantization target.**

## 2.2 KV Pattern Analysis

As established above, selecting a suitable partition is crucial for minimizing variance. We therefore analyze the K and V caches to examine whether they exhibit exploitable latent patterns that can guide the construction of pattern vectors for quantization.

### 2.2.1 Origins and Evolution of K Cache Patterns

Prior work identifies outlier distributions in the K cache (Liu et al., 2024b; Hooper et al., 2024), and we extend this line of evidence with a systematic robustness analysis (Appendix B), which shows that a fixed model's K cache maintains a stable structure attributable to internal linear mappings and nonlinear activations rather than any particular prompt. To probe this origin, we run an input–decoupling experiment: for each token, we compare the K cache distribution when propagating only the token embedding to that obtained from the full hidden state carrying context. As shown in Fig. 2, outlier channels already appear with embedding-only input, adding context chiefly inflates overall magnitude and dynamic range while leaving the structural pattern intact. The invariance of this pattern across inputs indicates that reliable pattern estimates can be obtained directly from the observed activations, without heavy dependence on corpus-specific calibration. We conclude:

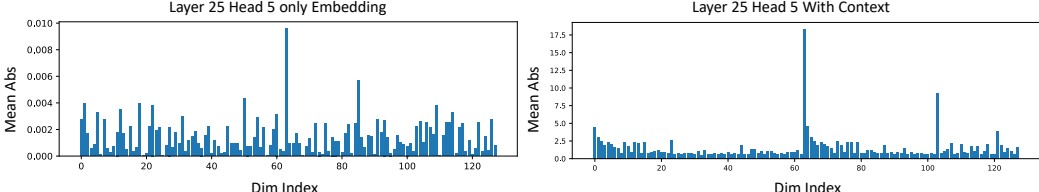

Figure 2: Channel-wise mean absolute value distributions. Left: embedding-only injection; Right: full-input injection. Outlier channels are already evident under embedding-only input, and the full input further enlarges the range and extremes. Additional figures are provided in Appendix C.

> **Insight 1**
>
> The stable structure in the K cache is primarily model-internal. Context mainly rescales values rather than altering the underlying structure.

Building on Insight 1, we analyze how the evolving context reshapes the K cache distribution during decoding. We sample K vectors along a single inference trajectory and visualize them per attention

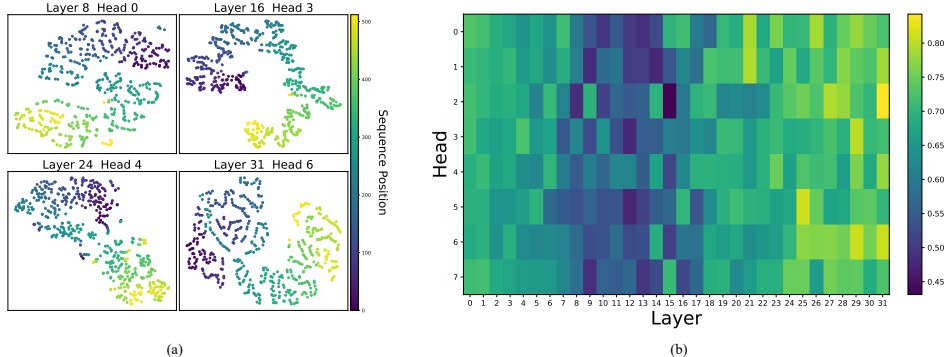

(a)                                                                                          (b)

Figure 3: (a) t-SNE visualization of the of K-cache distributions across attention heads along a single inference trajectory. (b) Illustration of the degree of alignment between V cache clusters and semantic categories. Additional figures are provided in Appendix C.

head using t-SNE. As shown in Fig. 3(a), the K distribution drifts smoothly across decoding steps rather than exhibiting abrupt jumps, and each head follows a distinct trajectory. This behavior is consistent with rotary positional embeddings, which inject relative position after Q and K are formed. Notably, although the marginal distribution evolves, short-range geometry remains stable: nearby tokens along the sequence tend to inhabit similar regions. This local consistency makes it natural to ground pattern estimates in the immediate neighborhood along the trajectory, where local similarity is highest. Hence,

> **Insight 2**
>
> Context and RoPE induce a gradual, head-specific evolution of the K distribution whose direction is difficult to predict.

### 2.2.2 ANALYSIS OF LATENT PATTERN IN V CACHE

In contrast to the K cache, the V cache shows neither pronounced outliers nor a broad dynamic range, so magnitude-only cues are uninformative. Because K does not appear in the output while V does, we instead rely on V's semantic content to uncover common structure. We therefore hypothesize a linkage to token semantics. To obtain a conservative estimate of semantic association, we proceed as follows: for each layer and head, we cluster V vectors using KMeans (McQueen, 1967). For tokens that appear multiple times, we compute their frequency distribution over clusters and define a consistency metric:

$$C_t = \frac{\max_k n_{t,k}}{\sum_k n_{t,k}} \tag{3}$$

We then aggregate $C_t$ across layers and heads to assess within-cluster cohesion. As shown in Fig. 3(b), shallow and deep layers exhibit strong alignment between tokens and cluster assignments, supporting our hypothesis. In the middle layers, the same token spreads across multiple clusters, indicating weaker coupling between V representations and semantics, which hampers the extraction of common structure. Therefore,

> **Insight 3**
>
> The V cache generally exhibits latent semantic patterns, with the association remaining strong in most layers and attenuating in some middle layers.

## 3 METHOD

In light of the previous analysis, we propose **PatternKV**, a residual quantization pipeline based on pattern alignment, as illustrated in Fig. 4. In the prefill stage, we select pattern vectors online via

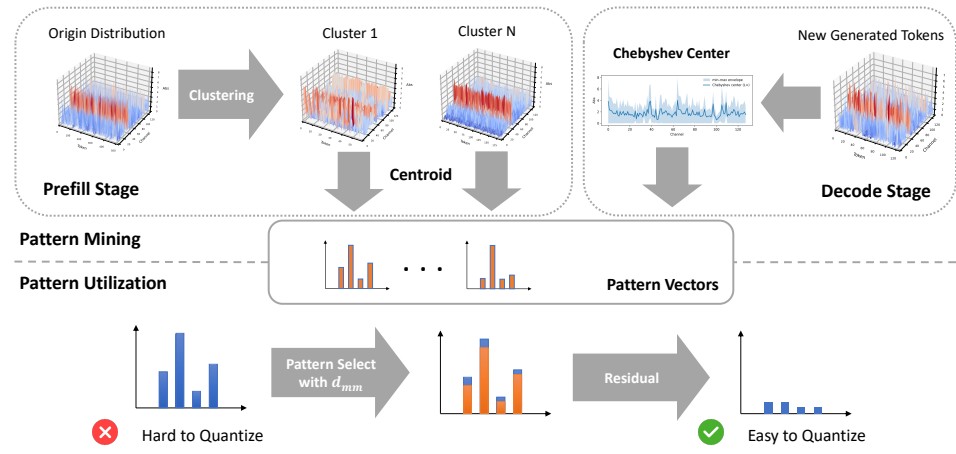

Figure 4: Overview of the PatternKV pipeline: pattern vectors are mined online, KV vectors are aligned to their nearest pattern, and only residuals are quantized.

clustering to minimize within-pattern variance (**Insight 1**). In the decode stage, we update the pattern vector to the Chebyshev center to adaptively track the distribution's gradual evolution (**Insight 2**). For pattern utilization, we assign each KV vector to a pattern using the min–max distance and quantize only the residual, which flattens the target distribution and contracts its dynamic range. For the V cache, where semantic alignment is weaker in intermediate layers (**Insight 3**), we further incorporate an adaptive threshold so that flattening provably incurs error no greater than raw quantization. Besides, we provide one theoretical guarantee for the method in Appendix D.

## 3.1 Pattern Mining

**Prefilling Stage** During the prefilling stage, we select and fix a set of pattern vectors so that the variance to be quantized reduces to $\mathbb{E}[\text{Var}(Z \mid M)]$. Our goal is therefore to minimize the within-pattern variance within the chosen partition, with the following optimization objective:

$$\min_{\mathcal{P}_h = P_1, \ldots, P_k} \sum_{j=1}^{k} \sum_{\boldsymbol{x_i} \in P_j} \left\| \boldsymbol{x_i} - \overline{\boldsymbol{x}}_{\boldsymbol{P_j}} \right\|_2^2 \tag{4}$$

Let $P_k$ denote the $k$-th pattern cluster. We optimize the objective using KMeans (McQueen, 1967) under the Euclidean metric and take the centroid of each cluster as its pattern vector. For the $h$-th attention head, the resulting set of pattern vectors is $\mathcal{M}_h = \{M_1, \ldots, M_k\}$. Since our objective coincides with the K-means objective, the partition returned at convergence is a local minimizer of the within-pattern variance.

**Decoding Stage** Guided by Insight 2, we update per-head pattern vectors during decoding to adaptively track the distribution's gradual evolution. Instead of arithmetic means, we use Chebyshev centers computed over each group of KV vectors, which minimize the local quantization range and provide stronger robustness to outliers, thereby aligning better with the asymmetric quantization objective. Specifically, we use the quantization group window $G_{\text{pattern}}$ to generate new pattern vectors. For the pattern vector $M_h^{new}$ of the $h$-th attention head within this window, we have:

$$M_{h,d}^{new} = \frac{1}{2} \left( \min_i X_{h,i,d} + \max_i X_{h,i,d} \right) \tag{5}$$

Here, $d$ indexes the dimension of the head, and $i$ indexes the $i$-th KV vector within the full-precision window. Once the $M_h^*$ is computed, it is merged into the existing pattern vertor set $\mathcal{M}_h$ for subsequent pattern matching and flattening.

## 3.2 PATTERN UTILIZATION

The objective of KV flattening is to minimize the quantization range. To achieve this, we replace direct quantization of raw vectors with residual quantization: each vector is first aligned to a pattern vector, and then only the residual is quantized, which yields a much flatter distribution.

Specifically, we adopt the min–max distance for pattern selection, defined for a vector $\boldsymbol{x}$ and a candidate pattern $\boldsymbol{m}$ as $d_{mm}(\boldsymbol{x}, \boldsymbol{m}) = \max_i(x_i - m_i) - \min_j(x_j - m_j)$.

During inference, for each KV vector we retrieve its nearest pattern under the $d_{\mathrm{mm}}$ metric. Concretely, the quantized target is the residual aligned to the nearest pattern:

$$M^\star = \underset{M \in \mathcal{M}_h}{argmin} \; d_{\mathrm{mm}}(X, M), \qquad R = X - M^\star \tag{6}$$

We record the index $k^\star$ together with the quantization parameters. During dequantization, we use this index to retrieve the corresponding pattern vector $M^\star$ and reconstruct the original KV representation by inverting the residualization step.

## 3.3 FLATTENING-SENSITIVE ADAPTIVE THRESHOLD FOR V PATTERN UTILIZATION

In mid layers, weak semantic associations can make flattening unreliable. To safeguard against this, we derive an adaptive threshold using a one-sided z-test, deciding whether to utilize the patterns. Define

$$D = \frac{1}{d} \sum_{i=1}^{d} \left( \varepsilon_{\mathrm{raw},i}^2 - \varepsilon_{\mathrm{flat},i}^2 \right) \tag{7}$$

where $\varepsilon_{(\cdot),i}$ denotes the error on dimension $i$ and $d$ is the head dimensionality. The null hypothesis is

$$H_0 : \; \mathbb{E}[D] \leq 0$$

, with significance level $\alpha$. Flattening is applied only when $H_0$ is rejected; otherwise, we revert to raw quantization. Under the high-resolution approximation, the following relationship for the quantization error of the V cache can be derived:

$$\mathbb{E}[D] = \frac{\Delta_{raw}^2 - \Delta_{flat}^2}{12}, \qquad \mathrm{Var}(D) = \frac{\Delta_{raw}^4 + \Delta_{flat}^4}{180\,d} \tag{8}$$

Here $\Delta_{(\cdot)} = \frac{R_{(\cdot)}}{2^n - 1}$ denotes the n-bit quantization step size. Because the head dimensionality $d$ in modern LLMs is typically large (e.g., 96, 128, 256), by the central limit theorem $D$ is approximately normal. Using the least favorable boundary $\mu = 0$, the one-sided $z$-test adopts the rejection region:

$$\frac{\mathbb{E}[D] - 0}{\sqrt{\mathrm{Var}(D)}} \geq z_{1-\alpha} \tag{9}$$

By substituting Eq. 8 and the definition of the quantization step size, and defining the contraction ratio as $\rho = R_{\mathrm{flat}}/R_{\mathrm{raw}}$, we obtain the key criterion:

$$1 - \rho^2 \; \geq \; \frac{2\,z_{1-\alpha}}{\sqrt{5\,d}} \; \sqrt{1 + \rho^4} \quad \Longleftrightarrow \quad \rho \; \leq \; \rho_*(d, \alpha) \tag{10}$$

Here $\rho_*(d, \alpha)$ denotes the solution to the equality in the left-hand criterion. Consequently, it suffices to compute online the quantization ranges before and after flattening, $R_{\mathrm{raw}}$ and $R_{\mathrm{flat}}$, and check whether $\rho = R_{\mathrm{flat}}/R_{\mathrm{raw}} \leq \rho_*(d, \alpha)$. If so, we conclude at confidence level $1 - \alpha$ that flattening yields a smaller quantization error for the current V vector.

## 3.4 SYSTEM OPTIMIZATION

**Prefill-stage pattern extraction** Since we need to extract different pattern vectors for different attention heads, we implement a fully GPU-parallel KMeans procedure to mitigate the inference latency introduced by clustering.

**Decode-stage KV reconstruction**  Restoring KV vectors during decoding incurs nontrivial overhead, so we introduce two customized CUDA kernels to reduce this cost. The first kernel **fuses** three steps—pattern-index restoration for the K cache, dequantization, and the QK matmul—into a single operator. The second kernel **fuses** the application of attention weights with quantized V, residual V, and pattern-index restoration into one operator. Together, these fused kernels greatly reduce the decode-stage latency compared with a pure PyTorch implementation.

## 4 EXPERIMENTS

### 4.1 SETTINGS

**Benchmarks**  As long contexts and test-time scaling commonly render the KV cache the dominant memory and bandwidth bottleneck during inference, we structure our evaluation into two categories. For the long-input setting, we use the full LongBench (Bai et al., 2024) benchmark, which offers multiple evaluation dimensions with task-specific metrics. LongBench details appear in Appendix E. For reasoning, we consider GSM8K (Cobbe et al., 2021), AIME (Balunović et al., 2025), and AMC (Li et al., 2024a). GSM8K probes the impact of quantization on chain-of-thought capability, and AIME and AMC evaluate performance under long chain-of-thought scenarios.

**Models**  To assess generalization, we evaluate two representative base model families: Llama (Dubey et al., 2024) and Qwen (Yang et al., 2024). Under the long-CoT setting, we employ Llama and Qwen variants distilled from DeepSeek-R1 (DeepSeek-AI et al., 2025) to enable longer chain-of-thought outputs.

**Baselines**  Because our method is an online algorithm that requires no offline calibration set, we compare it against online quantization baselines: KIVI (Liu et al., 2024b), ZipCache (He et al., 2024), SKVQ (Duanmu et al., 2024) and OTT (Su et al., 2025). Detailed experimental settings for the baseline methods are provided in Appendix F.

**Quantization Settings**  In all experiments of this section, we fix the number of pattern vectors at $|\mathcal{M}| = 32$ and set the quantization group for new pattern selection to $G_{\text{pattern}} = 128$. For quantization granularity, we use per-channel quantization for the K cache and per-token quantization for the V cache, matching the KIVI configuration. Since pre-RoPE recomputes rotary positional embeddings at every decoding step, we perform KV pattern selection after RoPE. All experiments were conducted on NVIDIA A100 GPUs with 40 GB of memory.

### 4.2 MAIN RESULTS

**Results on LongBench**  We evaluate on all 21 datasets of LongBench, focusing on two quantization precisions: INT2 and INT4. The 2-bit results in Table 1 demonstrate that our approach achieves robust gains over competitive baselines despite the extreme precision constraint. While some baselines achieve notable improvements on code-related tasks yet fail to generalize to other categories, our method provides stable and consistent improvements across task types. Results for INT4 setting are provided in Appendix G.

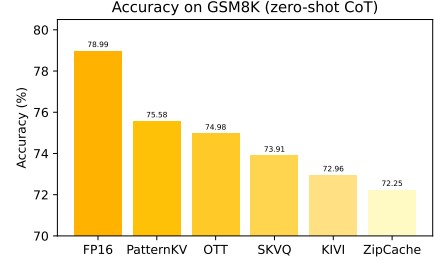

Figure 5: GSM8K accuracy under zero-shot CoT on Llama-3.1-8B-Instruct with an INT2 quantization setting.

**Results on Long-CoT Settings**  Test-time scaling improves LLM reasoning through depth-oriented expansion and breadth-oriented expansion, which yields long outputs and substantially increases KV cache usage. To this end, we evaluate models that can generate long chain-of-thought rationales on challenging mathematical benchmarks. For each problem, we generate eight independent responses and report $Avg@8$ (per-sample accuracy averaged over the eight responses) and $Maj@8$ (problem-level accuracy under majority voting across the eight responses). Table 2 reports the INT2 results: prior methods degrade markedly, whereas our method achieves an average 10% improvement. We also evaluate under the INT4 setting, detailed results are provided in Appendix H.

Table 1: Overall LongBench results at 2-bit precision. The best and second-best in every column are marked in **bold** and underline, respectively. See Appendix G for the 4-bit precision results.

| Model | Method | MQA | SQA | Summ. | Few-shot | Synth. | Code | Avg |
|---|---|---|---|---|---|---|---|---|
| Llama3.1-8B-Instruct | FP16 | 36.63 | 46.56 | 25.54 | 61.16 | 59.99 | 59.42 | 46.59 |
| | KIVI | 34.86 | 43.96 | 24.98 | 60.35 | 54.43 | 55.53 | 44.33 |
| | ZipCache | 32.65 | 40.52 | 24.02 | 59.86 | 47.44 | 60.91 | 42.49 |
| | SKVQ | 34.81 | 42.59 | 24.83 | 59.74 | 52.81 | 61.45 | 44.25 |
| | OTT | 34.34 | 43.41 | **25.19** | 59.64 | 55.45 | **62.48** | 44.84 |
| | PatternKV | **35.49** | **45.08** | 25.12 | **60.58** | **57.89** | 56.55 | **45.33** |
| Llama3.1-70B-Instruct | FP16 | 52.68 | 49.56 | 25.67 | 66.18 | 72.67 | 46.80 | 51.81 |
| | KIVI | 52.41 | 48.92 | **25.45** | 65.73 | 72.58 | 46.62 | 51.48 |
| | ZipCache | 36.98 | 45.44 | 23.28 | 58.57 | 67.92 | 58.37 | 46.55 |
| | SKVQ | - | - | - | - | - | - | - |
| | OTT | 40.72 | 47.36 | 24.74 | 60.05 | 68.50 | **59.97** | 48.43 |
| | PatternKV | **52.45** | **49.19** | 25.21 | **65.76** | **72.67** | 47.65 | **51.61** |
| Qwen2.5-7B-Instruct | FP16 | 38.03 | 45.40 | 23.37 | 59.85 | 58.83 | 62.84 | 46.13 |
| | KIVI | 35.77 | 42.73 | **22.80** | 58.13 | 51.50 | 56.25 | 43.08 |
| | PatternKV | **36.36** | **43.93** | 22.77 | **59.21** | **55.17** | 56.67 | **44.18** |

Table 2: Overall Results on the Long-CoT Benchmark at 2-bit precision. See Appendix H for the 4-bit precision results.

| Model | Method | AIME 25 | | AIME 24 | | AMC 24 | | AMC 23 | |
|---|---|---|---|---|---|---|---|---|---|
| | | Avg@8 | Maj@8 | Avg@8 | Maj@8 | Avg@8 | Maj@8 | Avg@8 | Maj@8 |
| Llama-8B | FP16 | 32.33 | 37.93 | 37.93 | 61.55 | 53.06 | 60.22 | 85.58 | 90.13 |
| | KIVI | 12.50 | 17.33 | 10.83 | 14.0 | 30.52 | **46.05** | 62.19 | 78.0 |
| | PatternKV | **17.50** | **27.17** | **16.25** | **21.33** | **34.44** | 42.11 | **63.44** | **83.13** |
| Qwen-7B | FP16 | 38.39 | 52.14 | 51.67 | 71.67 | 60.51 | 63.18 | 90.06 | 94.87 |
| | KIVI | 27.92 | 35.0 | **43.75** | **59.33** | 56.11 | 64.0 | 83.33 | 90.0 |
| | PatternKV | **30.42** | **41.33** | 42.92 | 53.67 | **57.22** | **65.89** | **84.06** | **90.26** |
| Qwen-14B | FP16 | 45.83 | 63.17 | 64.58 | 75.83 | 65.00 | 67.56 | 92.50 | 95.0 |
| | KIVI | **37.08** | **50.5** | 45.00 | 60.83 | 57.22 | 64.67 | 85.62 | **92.5** |
| | PatternKV | 35.42 | 46.67 | **47.92** | **68.16** | **62.22** | **67.78** | **88.12** | **92.5** |

**Results on GSM8K**  We use GSM8K to assess quantization in the non–long-text regime, adopting a zero-shot chain-of-thought paradigm. All results in Fig. 5 are obtained under the INT2 quantization setting. Our method reduces accuracy loss in the non–long-text setting. This suggests that preserving the fundamental patterns of KV vectors is critical for maintaining accuracy on reasoning-intensive tasks.

## 4.3 ABLATION STUDIES

We conduct two sets of ablation studies: the first evaluates the contribution of individual components, and the second examines the effect of the number of pattern vectors. Experiments are performed on Llama-3.1-8B-Instruct using LongBench and GSM8K.

**Components**  Table 3 shows that each component contributes positively to the overall method. Most notably, removing the adaptive threshold on the V cache leads to substantial performance

Table 3: Ablation on Components.

| Component | LongBench Avg | GSM8K |
|---|---|---|
| KIVI | 44.33 | 72.96 |
| PatternKV | 45.33 | 75.58 |
| w/o K Pattern | 44.53 | 73.91 |
| w/o V Pattern | 44.96 | 74.60 |
| w/o New Pattern | 45.37 | 75.49 |
| w/o V Threshold | 24.67 | 0.30 |

Table 4: Ablation on the number of patterns.

| $|\mathcal{M}|$ | LongBench Avg | GSM8K |
|---|---|---|
| KIVI | 44.33 | 72.96 |
| 2 | 44.57 | 73.72 |
| 4 | 44.92 | 75.26 |
| 8 | 44.92 | 75.94 |
| 16 | 45.28 | 75.20 |
| 32 | 45.33 | 75.58 |

degradation. This observation corroborates our earlier analysis: because semantic alignment on V varies across layers, a limited number of patterns cannot adequately cover its distribution, the nearest pattern to a given vector may thus be substantially biased, motivating a conservative rejection rule. Despite this, our approach maintains a high level of pattern utilization (about 75%). For more details, see Appendix I. We also find that leveraging patterns on K yields larger gains than on V, consistent with Hariri et al. (2025). Under low-bit settings, allocating greater quantization slack to K yields superior quantization benefits.

**The number of patterns** As shown in Table 4, quantization accuracy improves monotonically with the number of patterns. Notably, with $|\mathcal{M}| = 4$, we obtain roughly half of the total gains on LongBench and nearly all of the gains on GSM8K. This suggests a task-dependent choice of $|\mathcal{M}|$: long-context tasks benefit from a larger pattern budget to ensure robust coverage, whereas non–long-context tasks achieve comparable accuracy with a smaller number of patterns.

## 4.4 EFFICIENCY AND RESOURCE OVERHEAD ANALYSIS

We profile inference throughput and peak memory on an NVIDIA H20 (96 GB) GPU. The input length is fixed at 1024 and the output length at 256; batch sizes are $\{16, 32, 48, 64, 96, 128, 160\}$. The model is Llama-3.1-8B-Instruct. As shown in Fig. 6, compared with FP16 our method attains **1.4×** higher throughput and increases the single-GPU maximum batch size by **1.25×**.

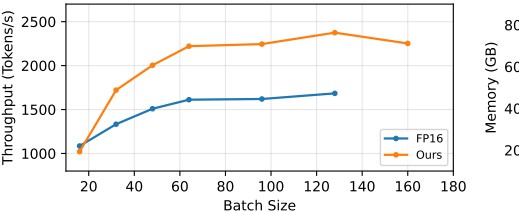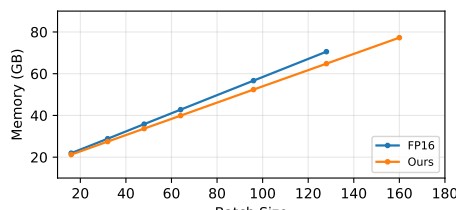

Figure 6: Comparison of throughput and memory footprint on Llama-3.1-8B-Instruct.

## 5 RELATED WORK

KV quantization has developed along three lines of work: (i) outlier-aware compression, where early systems Sheng et al. (2023) showed the feasibility of 4-bit KV but suffered at lower precision; Liu et al. (2024b) pushed to 2 bits with asymmetric quantization, and later methods Hooper et al. (2024); Duanmu et al. (2024); Su et al. (2025) mitigated outliers by separating dense and sparse components, constraining error drift, and exempting anomalous tokens; (ii) mixed precision and sensitivity adaptation, where evidence that keys are more fragile than values motivates Tao et al. (2025) to allocate higher precision to K, while He et al. (2024) adapts per-token bit widths to capture temporal importance; and (iii) KV sparsification and selective access, where Zhang et al. (2024) compress channels into compact codebooks and Kumar (2024) stack residual codebooks to approximate KV vectors at a reduced bitrate. Unlike prior work that partitions or approximates the raw KV distribution, our method explicitly flattens it. By contracting the dynamic range, we unlock greater quantization redundancy and preserve accuracy in low-bit settings.

Complementary to quantization, KV pruning targets redundancy by removing unimportant content before storage. Research follows two lines: (i) sequence-level token selection, where Xiao et al. (2024) retain recent tokens via sliding windows, Zhang et al. (2023); Liu et al. (2023) identify heavy hitters using attention scores, and Chitty-Venkata et al. (2025); Wang et al. (2025); Wu et al. (2024); Liu et al. (2024a); Li et al. (2024b) further improve saliency and cache stability via block eviction, one-shot top-k, soft voting, hashing, and representative snapshots; and (ii) structure-level compression, where Xu et al. (2025); Lv et al. (2025) prune low-value K/V channels and Tang et al. (2024) loads only query-relevant KV pages via query-aware metadata. Overall, historical KV caches are highly redundant, with importance concentrated in a small subset of tokens or channels.

## 6 CONCLUSION

We analyze common patterns in KV caches through a variance–decomposition perspective and introduce PatternKV, a lightweight quantization scheme that reshapes the KV distribution. By mining pattern vectors and quantizing residuals, PatternKV reduces intra-pattern variance and contracts the dynamic range, yielding flatter distributions and higher fidelity under low-bit settings. We establish theoretical support for the method and validate its effectiveness with extensive experiments, while also pointing toward more efficient implementations and system-level integration for broader deployment of LLMs.

## 7 ETHICS STATEMENT

All datasets used in this study are publicly available; no human subjects or annotators were involved. We confirm that our use is consistent with the datasets' licenses and research intent, and that no personally identifiable or harmful content is included. We cite all datasets and related works accordingly.

## 8 REPRODUCIBILITY STATEMENT

We take several steps to ensure reproducibility: we provide detailed information on the benchmarks and their usage, the complete parameter settings for all baseline methods as well as our method, and full hardware specifications.

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

## A    USE OF LLMs

We used large language models (LLMs) only as a general-purpose writing aid. LLMs did not contribute to research ideation, experiment design, implementation, analysis, or result interpretation, and no text was directly copied without human review. No proprietary or sensitive data were provided to LLMs. All technical content, claims, and conclusions are authored and verified by the authors.

## B    K CACHE PATTERN STABLE ANALYSIS

Prior studies have largely focused on outliers in K cache along a single trajectory, with limited evaluation of cross-trajectory consistency under different sampling paradigms. To address this, we build an evaluation set from GSM8K and run the model under two settings: parallel inference and multi-sample decoding. We then compute and compare mutual information for three cases: between tokens across different prefill runs, between tokens across distinct inference trajectories, and between different token positions within a single trajectory. Higher mutual information indicates greater common structure in K cache and stronger consistency, both across and within trajectories.

Table 5: Mutual Information of K Across Prefill Runs, Trajectories, and Token Positions

| Model | Random | Inter-Prefill | Inter-Sample | Inter-Token |
|---|---|---|---|---|
| Llama-3.1-8B-Instruct | | 0.1868 | 0.1771 | 0.1829 |
| Mistral-7B-Instruct-v0.3 | 0.0039 | 0.2067 | 0.2224 | 0.2291 |
| Qwen2.5-7B-Instruct | | 0.4169 | 0.4169 | 0.4121 |

From table 5, mutual information measured on the K-cache differs across model families; however, for any fixed model, the K-cache mutual information remains highly consistent across settings. Since the primary variation across inference paradigms lies in the composition of the presented context and the resulting trajectories, we arrive at the following observation: For any context, a given model's K-cache retains a nontrivial amount of stable structural information.

## C SUPPLEMENTARY FIGURES FOR INSIGHT 1 AND INSIGHT 2

**Settings** All preliminary experiments were conducted on the GSM8K dataset. To ensure fair comparison, we employed a unified zero-shot chain-of-thought (CoT) prompt template across all models and experimental conditions. This design eliminates variability due to prompt formatting and allows a cleaner assessment of the impact of the proposed method.

> **CoT Prompt**
>
> {Question}Please reason step by step, and put your final answer within \boxed{}.

We provide additional experimental observations that corroborate our insights. See Figs. 7 and 8.

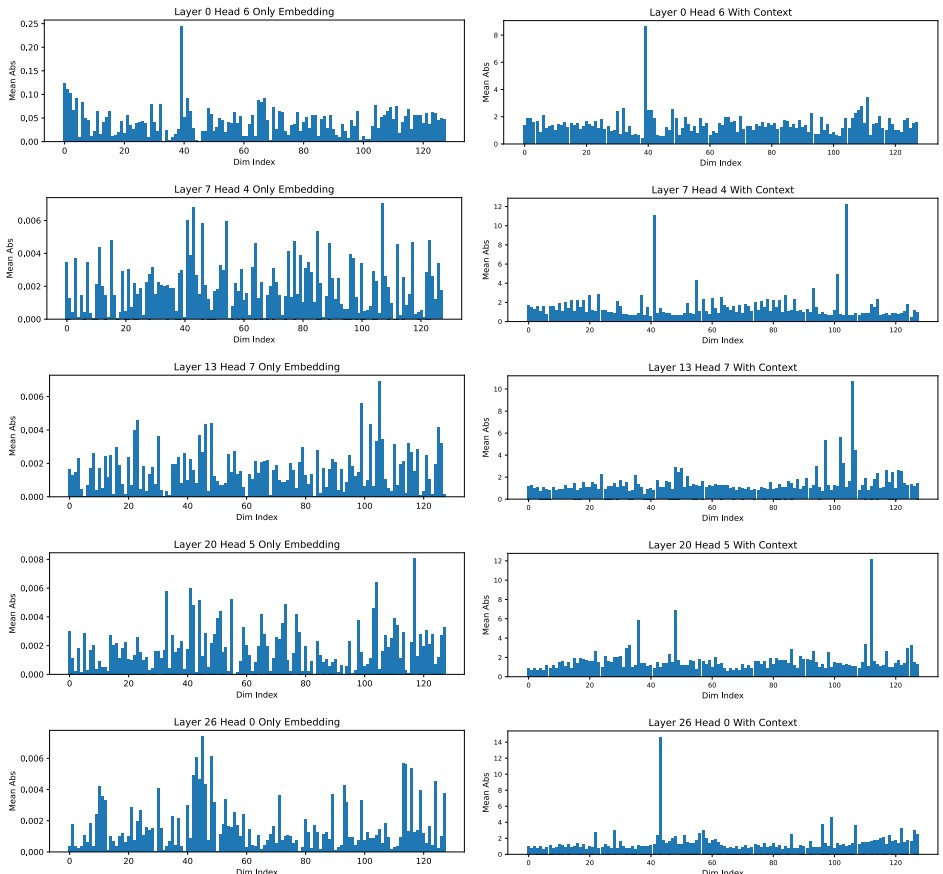

Figure 7: **Additional evidence for Insight 1.** We observe similar phenomena across different layers, supporting that the K-cache stable structure chiefly originates from the model.

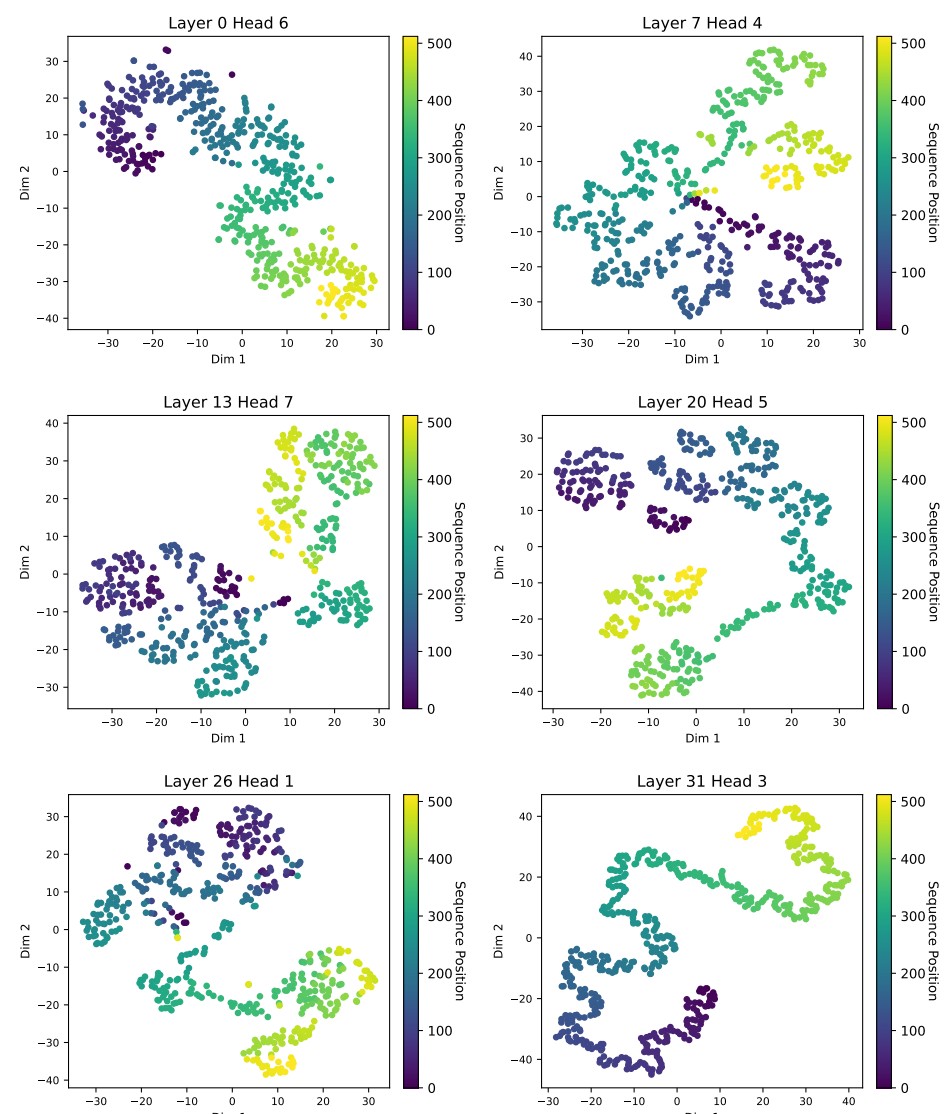

Figure 8: **Additional evidence for Insight 2.** The K cache shows layer- and head-specific evolution as the context grows over the decoding trajectory.

## D  ADDITIONAL PROOF

**Goal.** Show that for any bit-width $b \geq 1$ and any $\rho \in (0,1)$, there exists a finite pattern set $\mathcal{P}$ such that the residual scheme attains a uniform worst-case error bound satisfying

$$U_{\text{res}}^\star(b) \leq \rho \, U_{\text{raw}}^\star(b).$$

and this guarantee holds independently of the sequence length.

Let $X \in \{K, V\}$ denote the per-token key or value in $\mathbb{R}^d$. For any model input and position $t \geq 1$, write $X_t$ for the resulting vector. Assume bounded token embeddings and positional signals:

$$\max_w \|\text{emb}(w)\|_2 \leq M, \qquad \sup_{t \geq 1} \|\text{pos\_emb}(t)\|_2 \leq N. \tag{11}$$

Let $H_t^{(\ell)}$ be the hidden state at layer $\ell$. Denote by LN the normalization used in the block and by $\Psi$ the remainder of the block's mapping (attention + FFN + residual, etc.). We assume:

$$\|\Psi(h)\|_2 \leq L_\Psi \|h\|_2 + B_\Psi, \qquad \|\text{LN}(h)\|_2 \leq c_{\text{LN}} \|h\|_2 + b_{\text{LN}} \tag{12}$$

for constants $L_\Psi, B_\Psi, c_{\mathrm{LN}}, b_{\mathrm{LN}}$ that do not depend on sequence length or position. For the head's projection to $X$, write

$$X = W_X \, \mathrm{LN}\big(H^{(\ell)}\big), \qquad \|W_X\|_{2\to 2} = \sigma_X. \tag{13}$$

The input satisfies $H_t^{(0)} = \mathrm{emb}(w_t) + \mathrm{pos\_emb}(t)$ and hence $\sup_t \|H_t^{(0)}\|_2 \leq M + N$. Define $S_\ell := \sup_{t\geq 1} \|H_t^{(\ell)}\|_2$. Using the residual update and the linear hypotheses,

$$\|H_t^{(\ell+1)}\|_2 \leq \|H_t^{(\ell)}\|_2 + \|\Psi(\mathrm{LN}(H_t^{(\ell)}))\|_2 \leq a \, \|H_t^{(\ell)}\|_2 + b, \tag{14}$$

where $a := 1 + L_\Psi c_{\mathrm{LN}}$ and $b := L_\Psi b_{\mathrm{LN}} + B_\Psi$. Taking suprema over $t$ gives

$$S_{\ell+1} \leq a \, S_\ell + b, \qquad S_0 \leq M + N \quad \Rightarrow \quad S_\ell \leq a^\ell (M+N) + \frac{a^\ell - 1}{a - 1} \, b. \tag{15}$$

Therefore,

$$\sup_{t\geq 1} \|X_t\|_2 = \sup_t \|W_X \, \mathrm{LN}(H_t^{(\ell)})\|_2 \leq \sigma_X \big(c_{\mathrm{LN}} S_\ell + b_{\mathrm{LN}}\big) =: R_2 < \infty. \tag{16}$$

Let

$$\mathcal{S}_X := \{X_t : \text{ all inputs, all } t \geq 1\} \subset B_2(0, R_2) \subset \mathbb{R}^d. \tag{17}$$

In finite dimensions, bounded sets are totally bounded: for every $\varepsilon > 0$ there exists a finite $\varepsilon$-net $\mathcal{N}_\varepsilon$ in $\ell_\infty$ such that $\mathcal{S}_X \subset \bigcup_{p\in\mathcal{N}_\varepsilon} B_\infty(p, \varepsilon)$. Writing $R_\infty := \sup_{x\in\mathcal{S}_X} \|x\|_\infty \leq R_2$, a crude covering estimate is

$$N_\infty(\mathcal{S}_X, \varepsilon) \leq \big(1 + 2R_\infty/\varepsilon\big)^d. \tag{18}$$

Define the (standard) $\ell_\infty$ Chebyshev radius and center

$$R^\star := \inf_{c\in\mathbb{R}^d} \sup_{x\in\mathcal{S}_X} \|x - c\|_\infty, \qquad c^\star \in \arg\min_c \sup_{x\in\mathcal{S}_X} \|x - c\|_\infty, \tag{19}$$

and note that $w(x - c) := \max_i(x_i - c_i) - \min_i(x_i - c_i) \leq 2\|x - c\|_\infty$.

To compare worst-case bounds without unnecessary slack, introduce the *width–Chebyshev radius*

$$R_w^\star := \tfrac{1}{2} \inf_{c\in\mathbb{R}^d} \sup_{x\in\mathcal{S}_X} w(x - c), \qquad c_w^\star \in arg\min_c \sup_{x\in\mathcal{S}_X} w(x - c). \tag{20}$$

For non-symmetric uniform min–max quantization on a group of size $g$ and $b$ bits, the *optimal uniform worst-case bound (OUWB)* for the *direct* scheme is

$$U_{\mathrm{raw}}^\star(b) := \inf_c \sup_{x\in\mathcal{S}_X} \frac{\sqrt{g}}{2} \frac{w(x - c)}{2^b - 1} = \frac{\sqrt{g}}{2} \frac{2R_w^\star}{2^b - 1}. \tag{21}$$

Let $w(z) := \max_i z_i - \min_i z_i$. Fix any $\rho \in (0, 1)$ and set $\varepsilon = \rho R_w^\star$. By total boundedness, select a finite $\varepsilon$-net $\mathcal{P} = \{p_1, \ldots, p_K\}$ in $\ell_\infty$ covering $\mathcal{S}_X$. For any $x \in \mathcal{S}_X$, choose $p(x) \in \mathcal{P}$ with $\|x - p(x)\|_\infty \leq \varepsilon$. Then

$$w\big(x - p(x)\big) \leq 2\|x - p(x)\|_\infty \leq 2\rho R_w^\star, \qquad \text{hence} \quad \sup_x \frac{\sqrt{g}}{2} \frac{w(x - p(x))}{2^b - 1} \leq \rho \, U_{\mathrm{raw}}^\star(b). \tag{22}$$

Infimizing over finite $\mathcal{P}$ yields the residual OUWB

$$U_{\mathrm{res}}^\star(b) \leq \rho \, U_{\mathrm{raw}}^\star(b). \tag{23}$$

Consequently, for any $b \geq 1$ and $\rho \in (0, 1)$, there exists a finite pattern set $\mathcal{P}$ such that the residual scheme achieves an optimal uniform worst-case bound that is a $\rho$-fraction of the direct scheme's optimal uniform worst-case bound, independently of sequence length.

# E  DETAILED INFORMATION OF LONGBENCH

Following the LongBench official documentation, we categorize tasks into six types. The tasks and accompanying configurations for each category are listed in Table 6.

Table 6: LongBench Overview

| Task Type | Task | Metric | Avg. Length | Language | #Samples |
|---|---|---|---|---|---|
| Multi-document QA | HotpotQA | F1 | 9151 | English | 200 |
| | 2WikiMultihopQA | F1 | 4887 | English | 200 |
| | MuSiQue | F1 | 11214 | English | 200 |
| | DuReader | Rouge-L | 15768 | Chinese | 200 |
| | MultiFieldQA-zh | F1 | 6701 | Chinese | 200 |
| Single-document QA | MultiFieldQA-en | F1 | 4559 | English | 150 |
| | NarrativeQA | F1 | 18409 | English | 200 |
| | Qasper | F1 | 3619 | English | 200 |
| Summarization | GovReport | Rouge-L | 8734 | English | 200 |
| | QMSum | Rouge-L | 10614 | English | 200 |
| | MultiNews | Rouge-L | 2113 | English | 200 |
| | VCSUM | Rouge-L | 15380 | Chinese | 200 |
| Few-shot | TriviaQA | F1 | 8209 | English | 200 |
| | SAMSum | Rouge-L | 6258 | English | 200 |
| | TREC | Accuracy | 5177 | English | 200 |
| | LSHT | Accuracy | 22337 | Chinese | 200 |
| Synthetic Task | PassageRetrieval-en | Accuracy | 9289 | English | 200 |
| | PassageCount | Accuracy | 11141 | English | 200 |
| | PassageRetrieval-zh | Accuracy | 6745 | Chinese | 200 |
| Code | LCC | Edit Sim | 1235 | Python/C#/Java | 500 |
| | RepoBench-P | Edit Sim | 4206 | Python/Java | 500 |

## F  BASELINE SETTINGS

This section details the baseline configurations. For KIVI (Liu et al., 2024b), we set group_size = 128 and residual_size = 128. For ZipCache (He et al., 2024), we assign unimportant_ratio = 0.875 to both the K and V caches to approximately align the memory footprint. For SKVQ (Duanmu et al., 2024), we follow the official implementation with group_size = 128, channel-reorder count of 8, and clip_ratio = 0.92. For OTT (Su et al., 2025), we configure group_size = 128, residual_size = 32, sink_num = 3, and max_sink_num = 32.

A potential source of confusion is the fact that the residual size differs between KIVI and OTT. This discrepancy does not bias the comparison; on the contrary, it is introduced to make the comparison fairer. In KIVI, the quantization strategy on the K cache does not maintain a continuous sliding window: after each group is quantized, that group is immediately cleared from the residual buffer. In contrast, OTT maintains a fixed sliding window of length residual_size throughout the entire decoding process, so that the last residual_size tokens always remain in higher precision.

To keep the *effective* number of FP16 tokens preserved during decoding aligned across methods, we therefore set OTT's residual_size = 32. This choice ensures that the number of unquantized tokens retained by OTT at each decoding step is comparable, on average, to the number of FP16 tokens maintained by KIVI under its group-based quantization behavior.

## G  INT4 RESULTS ON LONGBENCH

In the 4-bit setting, we evaluate our method alongside baselines. As shown in Table 7, our method incurs only a 0.08% accuracy drop relative to FP16, which is nearly lossless.

Table 7: Overall results on LongBench at 4-bit setting. The best and second-best in every column are marked in **bold** and underline, respectively.

| Model | Method | MQA | SQA | Summ. | Few-shot | Synth. | Code | Avg |
|-------|--------|-----|-----|-------|----------|--------|------|-----|
| | FP16 | 36.63 | 46.56 | 25.54 | 61.16 | 59.99 | 59.42 | 46.59 |
| | KIVI | 36.63 | **46.69** | 25.64 | 61.25 | 57.77 | 59.48 | 46.34 |
| Llama3.1-8B-Instruct | ZipCache | - | - | - | - | - | - | - |
| | SKVQ | 35.39 | 44.15 | 25.23 | 59.70 | 58.46 | **63.79** | 45.75 |
| | OTT | 35.39 | 44.61 | **25.70** | 60.00 | **58.92** | 63.75 | 46.05 |
| | PatternKV | **36.78** | 46.59 | 25.50 | **61.29** | 58.42 | 59.31 | **46.41** |
| | FP16 | 52.68 | 49.56 | 25.67 | 66.18 | 72.67 | 46.80 | 51.81 |
| | KIVI | **53.09** | 49.58 | 25.68 | **66.16** | 72.67 | 46.80 | **51.89** |
| Llama3.1-70B-Instruct | ZipCache | - | - | - | - | - | - | - |
| | SKVQ | - | - | - | - | - | - | - |
| | OTT | 43.17 | 47.96 | 25.10 | 61.01 | 68.67 | **60.78** | 49.36 |
| | PatternKV | 52.66 | **49.70** | **25.80** | 66.12 | **72.83** | 46.86 | 51.87 |
| | FP16 | 38.03 | 45.40 | 23.37 | 59.85 | 58.83 | 62.84 | 46.13 |
| Qwen2.5-7B-instruct | KIVI | 37.71 | **45.58** | **23.46** | 59.88 | 58.50 | 62.53 | 46.05 |
| | PatternKV | **38.33** | 45.00 | 23.36 | **60.04** | **59.17** | 62.78 | **46.19** |

## H   INT4 RESULTS ON LONG-CoT SETTINGS

In the 4-bit setting, we evaluate our method against baselines; the results are shown in Table 8. Overall accuracy is substantially restored, although a residual gap remains. On benchmarks with larger degradation (e.g., AIME25), our method often recovers a substantial portion of the accuracy.

Table 8: Overall results on long-CoT Benchmark at 4-bit setting.

| Model | Method | AIME 25 | | AIME 24 | | AMC 24 | | AMC 23 | |
|-------|--------|---------|-------|---------|-------|--------|-------|--------|-------|
| | | Avg@8 | Maj@8 | Avg@8 | Maj@8 | Avg@8 | Maj@8 | Avg@8 | Maj@8 |
| | FP16 | 32.33 | 37.93 | 37.93 | 61.55 | 53.06 | 60.22 | 85.58 | 90.13 |
| Llama-8B | KIVI | 24.58 | 31.33 | 37.92 | 57.16 | 52.50 | 65.88 | 86.86 | 92.56 |
| | PatternKV | **27.50** | **37.0** | **38.75** | **59.33** | 50.83 | 58.22 | 85.31 | 91.13 |
| | FP16 | 38.39 | 52.14 | 51.67 | 71.67 | 60.51 | 63.18 | 90.06 | 94.87 |
| Qwen-7B | KIVI | 38.67 | 49.33 | 49.58 | 73.33 | 58.89 | 66.0 | 89.06 | 93.75 |
| | PatternKV | 38.33 | 46.33 | **52.08** | **69.5** | **60.28** | **68.0** | 88.44 | **95.0** |
| | FP16 | 45.83 | 63.17 | 64.58 | 75.83 | 65.00 | 67.56 | 92.50 | 95.0 |
| Qwen-14B | KIVI | 42.08 | 53.33 | 63.33 | 76.67 | 61.67 | 66.89 | 91.88 | 95.0 |
| | PatternKV | **45.83** | **64.17** | 62.50 | **76.67** | **61.94** | 64.89 | **92.81** | **95.0** |

## I   V PATTERN UTILIZATION RATE

As shown in Fig. 9 for *TriviaQA*, utilization remains high even under thresholding, implying the presence of latent semantic regularities in V cache.

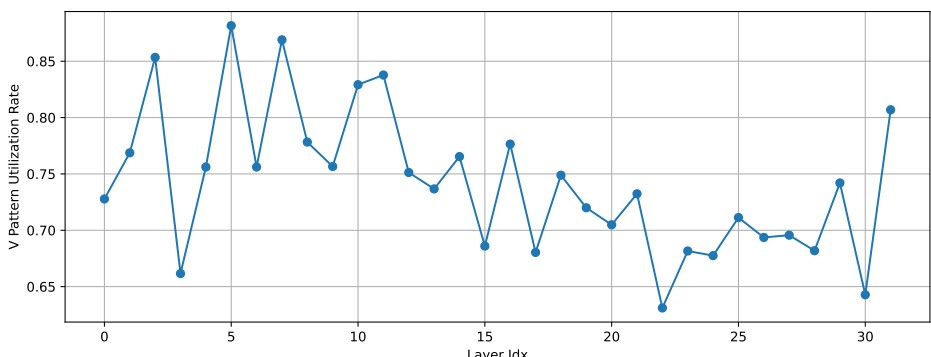

Figure 9: Visualization of V Pattern Utilization Rate on *TriviaQA*

Table 9: Component-level latency, throughput, and FLOPs associated with the PatternKV-specific procedures (batch size = 32). Latency and throughput are reported as relative change with respect to the baseline attention pipeline (negative throughput indicates a drop).

| Component | Latency | Throughput | GFLOPs |
|---|---|---|---|
| Pattern mining | $+6.60\%$ | $-6.10\%$ | 6.91 |
| Pattern selection | $+2.59\%$ | $-2.47\%$ | 2.07 |
| Chebyshev-center updates | $+3.11\%$ | $-2.94\%$ | 2.78 |

## J  ADDITIONAL EFFICIENCY ANALYSIS OF PATTERNKV

In this section, we provide a component-level breakdown of the computational overhead introduced by PatternKV, together with head-to-head comparisons against KIVI, ZipCache, OTT, and SKVQ in terms of latency, throughput, and peak GPU memory.

### J.1  COMPONENT-LEVEL BREAKDOWN

The additional computational components of PatternKV, beyond standard KV quantization, are:

- *Pattern mining (prefill).* K-means clustering over $|\mathcal{M}|$ patterns during the prefill stage.
- *Chebyshev-center updates (decode).* Non-iterative updates of pattern centers as decoding proceeds.
- *Pattern selection (decode).* At each decoding step, selecting the appropriate pattern via an index lookup and gathering the corresponding centroids.

Table 9 reports the overhead of these three PatternKV-specific procedures—pattern mining (prefill), pattern selection (decode), and Chebyshev-center updates (decode)—measured at batch size 32.

From Table 9, the dominant source of overhead is the pattern mining (clustering) step in the prefill stage, which is expected since K-means requires iterative updates to converge. In comparison, pattern selection and Chebyshev-center updates each contribute only about 2%–3% additional latency. Overall, these components introduce only modest extra cost relative to the full attention pipeline, while enabling the accuracy gains brought by PatternKV.

### J.2  HEAD-TO-HEAD LATENCY AND THROUGHPUT

Under identical hardware, model, quantization configuration, batch sizes, and sequence lengths, we compare end-to-end generation latency and throughput across FP16, KIVI, OTT, ZipCache, SKVQ, and PatternKV.

**Latency**  Table 10 summarizes the per-token latency (in seconds) for different batch sizes.

Table 10: End-to-end latency (s) under different batch sizes. "–" indicates that the corresponding configuration does not fit in GPU memory.

| Method | bz= 16 | bz= 32 | bz= 48 | bz= 64 | bz= 96 | bz= 128 | bz= 160 |
|---|---|---|---|---|---|---|---|
| FP16 | 11.29 | 18.39 | 24.46 | 30.57 | 45.78 | 58.70 | – |
| KIVI | 11.48 | 12.10 | 13.37 | 15.13 | 22.29 | 27.30 | 36.70 |
| OTT | 10.34 | 12.72 | 15.23 | 17.59 | 25.93 | 31.96 | – |
| ZipCache | 23.02 | 40.84 | 57.27 | 74.04 | 109.89 | 143.13 | – |
| SKVQ | 16.73 | 26.87 | 77.95 | 182.45 | 220.83 | – | – |
| PatternKV | 9.80 | 13.49 | 16.73 | 19.86 | 29.36 | 36.69 | 48.37 |

Table 11: End-to-end throughput (tokens/s) under different batch sizes.

| Method | bz= 16 | bz= 32 | bz= 48 | bz= 64 | bz= 96 | bz= 128 | bz= 160 |
|---|---|---|---|---|---|---|---|
| FP16 | 1089.33 | 1337.54 | 1508.60 | 1609.53 | 1612.52 | 1676.64 | – |
| KIVI | 1071.59 | 2032.88 | 2759.44 | 3251.66 | 3311.32 | 3605.49 | 3352.05 |
| OTT | 1189.93 | 1934.07 | 2423.10 | 2796.90 | 2846.63 | 3079.22 | – |
| ZipCache | 534.36 | 602.51 | 644.50 | 664.68 | 671.78 | 687.67 | – |
| SKVQ | 735.25 | 915.52 | 473.48 | 269.74 | 334.29 | – | – |
| PatternKV | 1255.05 | 1823.69 | 2206.27 | 2477.12 | 2514.12 | 2682.75 | 2543.68 |

**Throughput**   Table 11 reports the corresponding throughput (tokens per second).

From Tables 10 and 11, we observe that PatternKV incurs an increase in latency and a decrease in throughput compared with KIVI at larger batch sizes. This is expected, as PatternKV introduces additional components into the inference pipeline. At the same time, PatternKV delivers substantial accuracy improvements over KIVI (see main text), leading to a more favorable overall trade-off between efficiency and performance. Compared with ZipCache and SKVQ, PatternKV attains consistently better latency and throughput across all reported batch sizes, and it is competitive with OTT while being more accurate.

## J.3   PEAK MEMORY OVERHEAD

We next provide a fine-grained analysis of the memory overhead of PatternKV and validate it using empirical peak GPU memory measurements.

**Analytical memory estimates**   We maintain a separate set of centroids for each attention head, which introduces additional memory overhead. Let $L$ denote the number of layers, $H$ the number of KV heads, $B$ the batch size, and $S$ the sequence length (in tokens). For INT2 quantization, each stored value occupies 2 bytes.

*Prefill stage*   In the prefill stage, the number of pattern vectors is fixed regardless of the context length. The memory footprint of pattern vectors can be approximated as

$$M_{\text{pattern,prefill}} = 2LH \cdot 32 \cdot 2 \text{ bytes}$$

where the leading factor 2 accounts for both $K$ and $V$, and 32 is the number of pattern centroids per head. The KV cache memory usage is

$$M_{\text{KV}} = 2BLHS \cdot 2 \text{ bytes}$$

The relative memory fraction of pattern vectors during prefill is therefore

$$\frac{M_{\text{pattern,prefill}}}{M_{\text{KV}}} = \frac{32}{BS}$$

For a medium-length sequence with $S = 8192$ (approximately 8K) and $B = 1$, the pattern vectors account for only about $0.39\%$ of the KV memory. As the batch size increases, this fraction decreases further.

Table 12: Peak GPU memory consumption (GB) under different batch sizes.

| Method | bz= 16 | bz= 32 | bz= 48 | bz= 64 | bz= 96 | bz= 128 | bz= 160 |
|---|---|---|---|---|---|---|---|
| FP16 | 21.93 | 28.84 | 35.83 | 42.77 | 56.67 | 70.56 | – |
| KIVI | 21.19 | 27.39 | 33.57 | 39.77 | 52.16 | 64.55 | 76.95 |
| OTT | 24.07 | 31.13 | 38.15 | 45.20 | 59.31 | 73.41 | – |
| ZipCache | 23.10 | 29.21 | 35.35 | 41.43 | 53.61 | 65.84 | – |
| SKVQ | 24.68 | 32.37 | 40.06 | 47.75 | 63.13 | – | – |
| PatternKV | 21.23 | 27.48 | 33.69 | 39.90 | 52.39 | 64.81 | 77.28 |

*Decode stage* In the decode stage, we set $G_{\text{pattern}} = 128$, meaning that one pattern vector is generated for every 128 decoding steps. Analogously, the relative memory ratio of pattern vectors during decoding is

$$\frac{M_{\text{pattern,decode}}}{M_{\text{KV}}} = \frac{1}{BG_{\text{pattern}}}$$

This ratio is at most $0.78\%$ when $B = 1$, and again decreases as the batch size grows. In our implementation, the extra centroid tensors account for only about $0.42\%$ of the overall KV-cache memory footprint.

**Empirical peak GPU memory** Table 12 reports the peak GPU memory of each method during inference, which corroborates the above estimates.

Compared with the KIVI baseline, PatternKV introduces only about $0.42\%$ additional peak memory overhead while yielding a clear improvement in performance. This indicates that a very small extra storage cost is sufficient to obtain meaningful performance gains with PatternKV. In contrast, OTT, ZipCache, and SKVQ exhibit higher peak memory usage and less favorable scaling with the batch dimension, largely due to limitations in the publicly available implementations. Both KIVI and PatternKV maintain a well-controlled peak memory profile across the evaluated batch sizes.

## K K-DRIFT UNDER ALIBI POSITIONAL ENCODING

To further examine whether the observed drift of K patterns is primarily induced by RoPE, we additionally conduct pattern-evolution experiments on MPT-7B-Chat, a decoder-only model that adopts ALiBi positional encoding. Unlike RoPE-based models, the K cache in MPT-7B-Chat does not exhibit a clear progressive evolution with sequence length: tokens from early and late positions can share similar patterns, and abrupt changes may occur even between adjacent positions. Figure 10 visualizes the pattern assignments along the sequence dimension for representative attention heads. As shown in the figure, the pattern assignments under ALiBi remain relatively mixed across positions, without the smooth positional drift observed in the RoPE-based setting. These observations provide further evidence that the systematic drift of K patterns reported in the main paper is largely a consequence of RoPE, rather than an inherent property of attention weights or of the proposed pattern-mining procedure.

## L PATTERNKV UNDER MULTI-HEAD LATENT ATTENTION (MLA)

To assess how PatternKV extends to models with Multi-Head Latent Attention (MLA) and other non-standard attention architectures, we take DeepSeek-V2-Lite-Chat as a representative MLA model. Since, in MLA, RoPE is no longer applied to the entire K cache, we first verify whether the K cache in MLA still exhibits the key properties observed in GQA/MHA models.

Based on Figure 11 , we make the following observations:

- The RoPE-applied part of the K cache in MLA has substantially larger magnitudes than the non-RoPE part, which further suggests that RoPE is the primary source of the outlier distribution in the K cache.

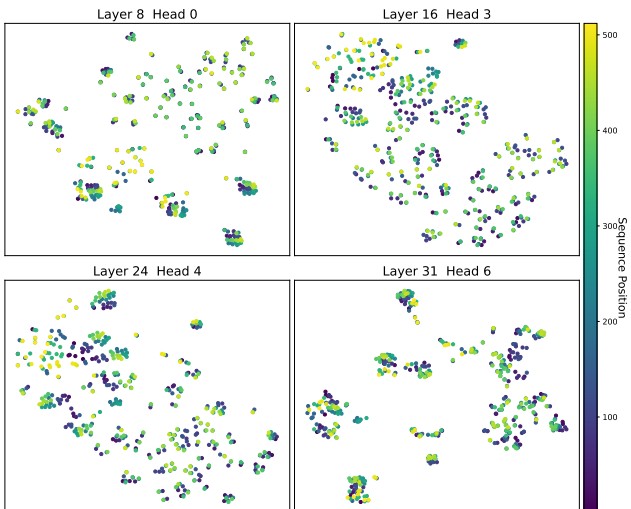

Figure 10: Pattern evolution of the K cache under ALiBi positional encoding on MPT-7B-Chat. Each row corresponds to an attention head, and colors denote discovered K patterns. Tokens from early and late positions can share similar patterns and abrupt pattern changes may occur between adjacent positions, in contrast to the smooth K-drift observed in RoPE-based models (see main text).

- The MLA K cache remains stable along the channel dimension, indicating that a small number of pattern vectors can still effectively cover its distribution.
- Along the sequence dimension, MLA exhibits the same gradual evolution behavior as GQA/MHA, which can be attributed to the RoPE information injected into the cache.

These findings indicate that PatternKV can be directly applied to the KV cache that is actually stored in MLA, without requiring any additional architectural adaptations. In future work, we plan to conduct more systematic MLA-based experiments to further validate these observations and to explore broader design choices for pattern-based KV compression in non-standard attention architectures.

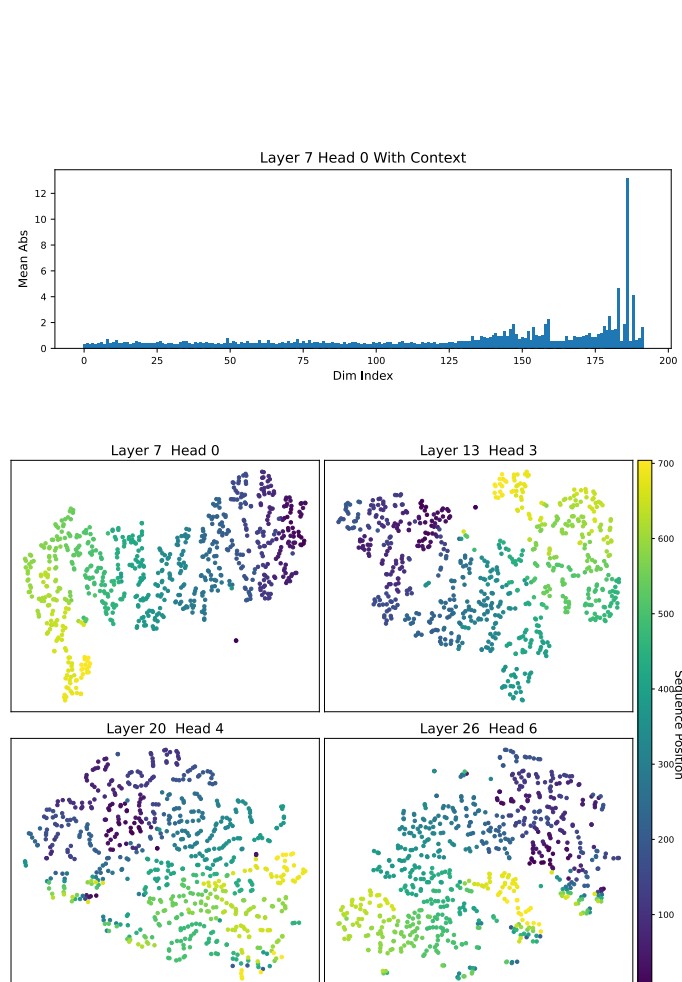

Figure 11: Pattern evolution of the K cache in DeepSeek-V2-Lite-Chat (MLA). The RoPE-applied subspace exhibits larger magnitudes and a gradual evolution along the sequence dimension, while remaining stable along channels, which mirrors the behavior observed in GQA/MHA and supports the applicability of PatternKV to MLA.

