# OpenReview forum: "PatternKV: Flattening KV Representation Expands Quantization Headroom"
_ICLR.cc/2026/Conference — Submitted to ICLR 2026_

### Official Review · Reviewer_pQ7J · 2025-10-30

**Soundness:** 3
**Presentation:** 3
**Contribution:** 3
**Rating:** 6
**Confidence:** 4

**Summary:**

This paper analyzes the distribution of the KV cache across layers and input tokens. Based on these insights, it presents PatternKV, a method that mines cluster centroid vectors to decompose the KV cache, making it more amenable to quantization. To validate its efficacy, the authors conduct evaluations on a range of long-context tasks using popular LLMs. Benchmark results demonstrate that the method achieves strong accuracy at 4-bit precision and consistent gains at 2-bit. Additional inference experiments demonstrate improved throughput and memory efficiency.

**Strengths:**

- The numerical analysis of the KV cache distribution is insightful. The theoretical foundation, based on variance decomposition, effectively motivates and drives the design of PatternKV.
- The accuray experimental evaluation is comprehensive, demonstrating the method's strong performance across a variety of scenarios, models, and benchmarks.

**Weaknesses:**

- The performance analysis of throughput and memory footprint is conducted on only a single model (Llama-3.1-8B) under a specific context length. To strengthen the claims, it would be beneficial to extend these efficiency tests to more models and a wider range of scenarios, such as longer input (e.g., 16k, 32k tokens).
- The paper lacks a detailed analysis of the computational overhead introduced by the online pattern mining, matching, and update processes. Quantifying this overhead would provide a more complete picture of the method's efficiency.

**Questions:**

- How would this method be applied, or how would it perform, in models that use Multi-Head Latent Attention (MLA) or other non-standard attention architectures?
- What is the memory overhead associated with storing the pattern vectors, and how does this scale with the number of layers, attention heads, and the chosen pattern set size?

---

> ### Author Response · Authors · 2025-11-23
> **Response to Reviewer pQ7J (Part-1)**
>
> Dear Reviewer pQ7J,
>
> Thank you for carefully reading our paper and for your valuable comments and insightful questions. Below, we provide detailed responses to each of the question you raised.
>
> > The performance analysis of throughput and memory footprint is conducted on only a single model (Llama-3.1-8B) under a specific context length. To strengthen the claims, it would be beneficial to extend these efficiency tests to more models and a wider range of scenarios, such as longer input (e.g., 16k, 32k tokens).
>
> We thank the reviewer for this helpful comment. We agree that evaluating only a single model and context length is limiting, and we have extended our analysis accordingly.
>
> **MHA efficiency experiment**
>
> To address this concern, we additionally conducted efficiency experiments on Llama-2-7B, which uses a standard MHA setup and is also the backbone used in the efficiency evaluations of both KIVI and OTT. The results are shown below.
>
> **Peak Memory**
>
> | Method    | bz=16 | bz=32 | bz=48 | bz=64 | bz=96 | bz=128 | bz=160 | bz=192 |
> |-----------|-------|-------|-------|-------|-------|--------|--------|--------|
> | FP16      | 23.64 | 34.69 | 45.75 | 56.80 | -     | -      | -      | -      |
> | PatternKV | 17.14 | 21.66 | 26.18 | 30.65 | 39.68 | 48.69  | 57.72  | 66.73  |
>
> **Latency**
> | Method    | bz=16 | bz=32 | bz=48 | bz=64 | bz=96 | bz=128 | bz=160 | bz=192 |
> |-----------|-------|-------|-------|-------|-------|--------|--------|--------|
> | FP16      | 17.81 | 32.91 | 44.20 | 57.61 | -     | -      | -      | -      |
> | PatternKV | 12.76 | 21.33 | 26.56 | 33.64 | 49.12 | 63.61  | 83.68  | 94.87  |
>
> **Throughput**
>
> | Method    | bz=16  | bz=32   | bz=48   | bz=64   | bz=96    | bz=128   | bz=160   | bz=192   |
> |-----------|-------|-------|-------|-------|-------|--------|--------|--------|
> | FP16      | 1149.74 | 1244.49 | 1389.80 | 1421.95 | -        | -        | -        | -        |
> | PatternKV | 1604.12 | 1919.47 | 2312.74 | 2435.15 | 2501.58  | 2575.51  | 2447.24  | 2590.38  |
>
> Across all evaluation settings on Llama-2-7B, PatternKV consistently demonstrates substantial efficiency gains. It reduces peak memory usage by **28%–46%**, accelerates inference latency by **28%–42%** (corresponding to a **1.4×–1.7×** end-to-end speedup), and improves throughput by **40%–71%**, with the throughput advantage continuing to grow as the batch size increases. These results clearly show that PatternKV provides strong efficiency benefits in the standard MHA architecture.
>
>
> **Longer sequences efficiency experiment**
>
> We also report the impact of different input lengths on inference latency and throughput under **batch size = 4** on the original Llama-3.1-8B setup.
>
> **Latency**
>
> | Method     | l=1k | l=2k | l=4k | l=8k | l=16k | l=32k |
> |------------|------|------|------|------|-------|-------|
> | FP16       | 1.78 | 2.12 | 3.25 | 5.55 | 10.42 | 21.22 |
> | PatternKV  | 3.68 | 3.00 | 3.53 | 4.87 |  8.63 | 19.20 |
>
> **Throughput**
>
> | Method     | l=1k   | l=2k   | l=4k   | l=8k   | l=16k  | l=32k  |
> |------------|--------|--------|--------|--------|--------|--------|
> | FP16       |1291.44 |2047.58 |2599.74 |2998.16 |3168.79 |3091.99 |
> | PatternKV  | 626.89 |1448.82 |2388.34 |3417.39 |3826.48 |3416.63 |
>
> For short contexts, the one-time cost of KMeans-based pattern mining dominates, so PatternKV is slightly slower than FP16. As the input length grows, this cost is amortized and the benefit of KV-cache compression increasingly dominates: **for 8k–32k tokens, PatternKV achieves both lower latency and higher throughput than FP16**. This confirms that our method remains efficient even in the long-context regime.
>
> In summary, by (i) adding experiments on **Llama-2-7B (MHA)** and (ii) extending the **context length** to up to **32k tokens**, we demonstrate that PatternKV delivers robust efficiency gains across **different architectures (GQA vs. MHA)** and **a wide range of practical input lengths**, thereby strengthening the generality of our efficiency claims.

---

> ### Author Response · Authors · 2025-11-23
> **Response to Reviewer pQ7J (Part-2)**
>
> > The paper lacks a detailed analysis of the computational overhead introduced by the online pattern mining, matching, and update processes. Quantifying this overhead would provide a more complete picture of the method's efficiency.
>
> We thank the reviewer for the helpful suggestion. Below we explicitly discuss the overheads of **clustering**, **pattern selection**, and **Chebyshev updates**, covering both computation and memory.
>
> **Component breakdown**
>
> In PatternKV, the main additional computational components beyond standard KV-quantization are:
>
> - **Pattern mining (prefill)** – K-means over |M| patterns.
> - **Chebyshev-center updates (decode)** – non-iterative updates of pattern centers as decoding proceeds.
> - **Pattern selection (decode)** – at each step, selecting the appropriate pattern via an index lookup and gathering the corresponding centroids.
>
> We report below the computational overhead of the three PatternKV-specific procedures—pattern mining (prefill stage), pattern selection (decode stage), and Chebyshev-center updates (decode stage). The results are measured at batch size = 32.
>
> | Component                 | Latency | Throughput  | GFLOPS |
> |---------------------------|---------|-------------|--------|
> | Pattern Mining            | +6.60%  | -6.10%      | 6.91   |
> | Pattern Selection         | +2.59%  | -2.47%      | 2.07   |
> | Chebyshev-center updates  | +3.11%  | -2.94%      | 2.78   |
>
> From these results, we see that the dominant source of overhead is the **pattern mining (clustering)** step in the prefill stage, which is expected since KMeans requires iterative updates to converge. In comparison, **pattern selection** and **Chebyshev-center updates** each contribute only about **2–3%** additional latency. Overall, these components together introduce only modest extra cost relative to the full attention pipeline, while enabling the accuracy gains brought by PatternKV.
>
>
> Overall, PatternKV introduces only modest performance overhead while yielding substantial accuracy gains.
>
> > How would this method be applied, or how would it perform, in models that use Multi-Head Latent Attention (MLA) or other non-standard attention architectures?
>
> This is indeed a very insightful question. We select Deepseek-V2-Lite-Chat as a representative MLA model. Since, in MLA, RoPE is no longer applied to the entire K cache, we first need to verify whether the K cache in MLA still exhibits similar properties to those observed in GQA/MHA. All experimental visualizations are provided in the Appendix. Based on these observations, we find that:
> - The RoPE-applied part of the K cache in MLA has much larger magnitudes than the non-RoPE part, which further suggests that RoPE is the primary source of the outlier distribution in the K cache.
> - The MLA K cache remains stable along the channel dimension, indicating that a small number of pattern vectors can still effectively cover its distribution.
> - Along the sequence dimension, MLA exhibits the same gradual evolution behavior as GQA/MHA, which can be attributed to the RoPE information injected into the cache.
>
> Therefore, our method can be directly applied to the KV cache actually stored in MLA without any additional adaptations. In future work, we plan to conduct more systematic MLA-based experiments to further validate these findings and explore broader design choices.

---

> ### Author Response · Authors · 2025-11-23
> **Response to Reviewer pQ7J (Part-3)**
>
> > What is the memory overhead associated with storing the pattern vectors, and how does this scale with the number of layers, attention heads, and the chosen pattern set size?
>
>
> Thank you for pointing this out, it is indeed an important question. We analyze the **memory overhead** of storing pattern centroids for clustering, pattern selection, and Chebyshev updates. Maintaining a separate set of centroids for each attention head introduces additional memory overhead, but in our implementation these extra centroid tensors account for only **0.42%** of the overall KV-cache memory footprint.
>
> * **Prefill stage.** In the prefill stage, regardless of the context length, the number of pattern vectors is fixed. We can roughly estimate the memory footprint as $ 2 \times  num\\_layer  \times num\\_kv\\_head \times 32(Pattern\\_size) \times 2 Byte $.
>   For the KV cache, the memory usage is
>   $2 \times batch\\_size \times num\\_layer \times num\\_kv\\_head \times num\\_seq \times 2\ Byte$.
>   This implies that the relative memory fraction of pattern vectors is
>   $32(Pattern\\_size) / (batch\\_size \times seq\\_num )$.
>   For a medium-length sequence with $seq\\_num = 8K $ and $ batch\\_size = 1$, the pattern vectors account for only about **0.39%** of the KV memory. As the batch size increases, this fraction becomes even smaller.
>
> * **Decode stage.** In the decode stage, we set $G_{pattern} = 128$, meaning that one pattern vector is generated for every 128 decoding steps. Analogously, the relative memory ratio of pattern vectors is
>  $1 / (batch\\_size \times 128)$, which is at most **0.78%** when $batch\\_size = 1$; again, this ratio further decreases as the batch size grows.
>
> We also report the peak GPU memory usage of PatternKV and other baseline methods during actual inference to empirically validate these overhead estimates.
>
> | Method     | bz=16 | bz=32 | bz=48 | bz=64 | bz=96 | bz=128 | bz=160 |
> |-----------|-------|-------|-------|-------|-------|--------|--------|
> | FP16      | 21.93 | 28.84 | 35.83 | 42.77 | 56.67 | 70.56  | -      |
> | KIVI      | 21.19 | 27.39 | 33.57 | 39.77 | 52.16 | 64.55  | 76.95  |
> | OTT       | 24.07 | 31.13 | 38.15 | 45.20 | 59.31 | 73.41  | -      |
> | Zipcache  | 23.10 | 29.21 | 35.35 | 41.43 | 53.61 | 65.84  | -      |
> | SKVQ      | 24.68 | 32.37 | 40.06 | 47.75 | 63.13 | -      | -      |
> | PatternKV | 21.23 | 27.48 | 33.69 | 39.90 | 52.39 | 64.81  | 77.28  |
>
> Compared to the KIVI baseline, our approach introduces only **0.42%** additional peak memory overhead while yielding a clear improvement in performance. This indicates that a very small extra storage cost for clustering centroids and pattern vectors is sufficient to obtain meaningful performance gains with PatternKV.
>
> We have incorporated the above component-wise analysis and efficiency experiments into the appendix.

---

### Official Review · Reviewer_2ewL · 2025-10-30

**Soundness:** 3
**Presentation:** 3
**Contribution:** 2
**Rating:** 4
**Confidence:** 5

**Summary:**

This paper proposes PatternKV, a pattern-aligned residual quantization scheme: (1) mining representative pattern vectors online, (2) aligning each KV vector to its nearest pattern, and (3) quantizing only the residual. The first 2 steps (flattening KV representation) benefits the third step (quantization) by narrowing the quantization range; in other words, the fidelity of low-bit KV quantization can be improved.

**Strengths:**

1. PatternKV, based on (1) mining representative pattern vectors online, (2) aligning each KV vector to its nearest pattern, and (3) quantizing only the residual, differs fundamentally from prior work focusing on outlier mitigation. Outlier mitigation suffers possible performance loss with low-bit quantization, while PatternKV is okay (mostly outperforms) with low-bit quantization.
2. 1.4x throughput increase seems promising (if the baseline setting is fair -- please also see "Questions" below).

**Weaknesses:**

1. The overheads for clustering, pattern selection, and Chebyshev updates may be significant and should be discussed.
2. V pattern utilization rate may affect the efficacy of PatternKV -- this is discussed in Section 3.3 but not thoroughly (and experimentally) analyzed.
3. Lack of direct comparisons of speed (latency or throughout) and memory consumption against related works. Comparing to the FP16 case is indirect and does not seem fair.
4. The font size of figures, especially Figure 1 - Figure 3, is way to small. I can barely see the words.

**Questions:**

My questions and suggestions are basically from "Weaknesses" as aforementioned.
1. From Weakness 1: Please discuss the overheads for clustering, pattern selection, and Chebyshev updates.
2. From Weakness 2: Please thoroughly and experimentally analyze the effect of V pattern utilization rate on PatternKV's efficacy.
3. From Weakness 3: Please use KIVI (or representative counterpart) with INT2/INT4 quantization as the baseline(s) for more fair comparisons of throughput and memory consumption. Comparing to the FP16 case is indirect and does not seem fair.

---

> ### Author Response · Authors · 2025-11-23
> **Response to Reviewer 2ewL (Part-1)**
>
> Dear Reviewer 2ewL,
>
> We sincerely appreciate your assessment of our work and your valuable suggestions. They will help us further improve the paper. Below, we address each of your questions in detail.
>
> > From Weakness 1: Please discuss the overheads for clustering, pattern selection, and Chebyshev updates.
>
>
> We thank the reviewer for the helpful suggestion. Below we explicitly discuss the overheads of **clustering**, **pattern selection**, and **Chebyshev updates**, covering both computation and memory.
>
> **Component breakdown**
>
> In PatternKV, the main additional computational components beyond standard KV-quantization are:
>
> - **Pattern mining (prefill)** – K-means over |M| patterns.
> - **Chebyshev-center updates (decode)** – non-iterative updates of pattern centers as decoding proceeds.
> - **Pattern selection (decode)** – at each step, selecting the appropriate pattern via an index lookup and gathering the corresponding centroids.
>
> We report below the computational overhead of the three PatternKV-specific procedures—pattern mining (prefill stage), pattern selection (decode stage), and Chebyshev-center updates (decode stage). The results are measured at batch size = 32.
>
> | Component                 | Latency | Throughput  | GFLOPS |
> |---------------------------|---------|-------------|--------|
> | Pattern Mining            | +6.60%  | -6.10%      | 6.91   |
> | Pattern Selection         | +2.59%  | -2.47%      | 2.07   |
> | Chebyshev-center updates  | +3.11%  | -2.94%      | 2.78   |
>
> From these results, we see that the dominant source of overhead is the **pattern mining (clustering)** step in the prefill stage, which is expected since KMeans requires iterative updates to converge. In comparison, **pattern selection** and **Chebyshev-center updates** each contribute only about **2–3%** additional latency. Overall, these components together introduce only modest extra cost relative to the full attention pipeline, while enabling the accuracy gains brought by PatternKV.
>
> Overall, PatternKV introduces only modest performance overhead while yielding substantial accuracy gains.
>
> **Peak-memory overhead**
>
> While maintaining a separate set of centroids for each attention head may introduce additional memory overhead, our detailed analysis shows that, in our implementation, these extra centroid tensors account for only **0.42%** of the overall KV-cache memory footprint. The detailed breakdown is provided below.
>
> * **Prefill stage.** In the prefill stage, regardless of the context length, the number of pattern vectors is fixed. We can roughly estimate the memory footprint as $ 2 \times  num\\_layer  \times num\\_kv\\_head \times 32 \times 2 Byte $.
>   For the KV cache, the memory usage is
>   $2 \times batch\\_size \times num\\_layer \times num\\_kv\\_head \times num\\_seq \times 2\ Byte$.
>   This implies that the relative memory fraction of pattern vectors is
>   $32 / (batch\\_size \times seq\\_num )$.
>   For a medium-length sequence with $seq\\_num = 8K $ and $ batch\\_size = 1$, the pattern vectors account for only about **0.39%** of the KV memory. As the batch size increases, this fraction becomes even smaller.
>
> * **Decode stage.** In the decode stage, we set $G_{pattern} = 128$, meaning that one pattern vector is generated for every 128 decoding steps. Analogously, the relative memory ratio of pattern vectors is
>  $1 / (batch\\_size \times 128)$, which is at most **0.78%** when $batch\\_size = 1$; again, this ratio further decreases as the batch size grows.
>
> We also report the peak GPU memory usage of PatternKV and other baseline methods during actual inference to empirically validate these overhead estimates.
>
> | Method     | bz=16 | bz=32 | bz=48 | bz=64 | bz=96 | bz=128 | bz=160 |
> |-----------|-------|-------|-------|-------|-------|--------|--------|
> | FP16      | 21.93 | 28.84 | 35.83 | 42.77 | 56.67 | 70.56  | -      |
> | KIVI      | 21.19 | 27.39 | 33.57 | 39.77 | 52.16 | 64.55  | 76.95  |
> | OTT       | 24.07 | 31.13 | 38.15 | 45.20 | 59.31 | 73.41  | -      |
> | Zipcache  | 23.10 | 29.21 | 35.35 | 41.43 | 53.61 | 65.84  | -      |
> | SKVQ      | 24.68 | 32.37 | 40.06 | 47.75 | 63.13 | -      | -      |
> | PatternKV | 21.23 | 27.48 | 33.69 | 39.90 | 52.39 | 64.81  | 77.28  |
>
> Compared to the KIVI baseline, our approach introduces only **0.42%** additional peak memory overhead while yielding a clear improvement in performance. This indicates that a very small extra storage cost for clustering centroids and pattern vectors is sufficient to obtain meaningful performance gains with PatternKV.

---

> ### Author Response · Authors · 2025-11-23
> **Response to Reviewer 2ewL (Part-2)**
>
> > From Weakness 2: Please thoroughly and experimentally analyze the effect of V pattern utilization rate on PatternKV's efficacy.
>
> This is an excellent and thought-provoking question. Since we cannot directly control the utilization rate of the V patterns, we instead adjust it by varying the confidence level $\alpha$ associated with the V patterns. The experimental results are summarized in the table below.
>
> | Metric                      | $\alpha$=0.99  | $\alpha$=0.95  | $\alpha$=0.90  | $\alpha$=0.85  | $\alpha$=0.80  | $\alpha$=0.60  | $\alpha$=0.40  | $\alpha$=0.20  | non-threshold |
> |-----------------------------|-------|-------|-------|-------|-------|-------|-------|-------|---------------|
> | LongBench                   | 45.36 | 45.33 | 45.31 | 45.35 | 45.32 | 44.19 | 40.03 | 31.84 | 24.67         |
> | GSM8K                       | 75.43 | 75.58 | 76.64 | 76.49 | 76.87 | 72.13 | 47.55 | 10.01 | 0.30          |
> | V pattern utilization rate (%) | 91.38 | 92.90 | 93.74 | 94.31 | 94.72 | 95.71 | 96.42 | 97.17 | 100           |
>
> On GSM8K, we empirically observe that accuracy **first increases and then decreases** as the fraction of V cache represented by pattern vectors grows, indicating that pattern utilization should be kept within a reasonable range. To better understand this phenomenon, we further examine how different confidence thresholds affect the fraction of V cache positions assigned to pattern vectors. The results indicate that even under a high confidence threshold, at least 90% of the V cache can still be represented using pattern vectors, which further supports the existence of underlying semantic patterns in the V cache. At the same time, we observe that under a low confidence threshold, around 3% of V cache entries do not pass the hypothesis test; forcing residual sharing on these tokens leads to a substantial performance drop. This confirms that our hypothesis-testing mechanism effectively isolates V cache positions that do not exhibit stable patterns, and quantizing them with the original method helps avoid amplified quantization error.
>
> > From Weakness 3: Please use KIVI (or representative counterpart) with INT2/INT4 quantization as the baseline(s) for more fair comparisons of throughput and memory consumption. Comparing to the FP16 case is indirect and does not seem fair.
>
> We fully agree with the reviewer on the importance of evaluating runtime overhead in practical inference scenarios. However, it is worth noting that among existing KV-quantization approaches, KIVI achieves strong latency and throughput largely due to its simple design and highly optimized implementation—**yet this simplicity also imposes an upper bound on its achievable accuracy**.
>
> Methods that focus on improving accuracy, including ours and several other baselines, inevitably introduce some system-level costs: higher-quality quantization may slightly reduce throughput, increase latency, or add modest memory overhead. In our case, PatternKV turns a very small additional cost into clear accuracy improvements, leading to a more favorable balance between model quality and resource usage.
>
> Below, we present the latency and throughput comparisons between our approach and the baseline methods.
>
> **Latency**
>
> | Method     | bz=16 | bz=32 | bz=48 | bz=64 | bz=96 | bz=128 | bz=160 |
> |------------|-------|-------|-------|-------|-------|--------|--------|
> | FP16       | 11.29 | 18.39 | 24.46 | 30.57 | 45.78 | 58.70  | -      |
> | KIVI       | 11.48 | 12.10 | 13.37 | 15.13 | 22.29 | 27.30  | 36.70  |
> | OTT        | 10.34 | 12.72 | 15.23 | 17.59 | 25.93 | 31.96  | -      |
> | ZipCache   | 23.02 | 40.84 | 57.27 | 74.04 |109.89 |143.13  | -      |
> | SKVQ       | 16.73 | 26.87 | 77.95 |182.45 |220.83 | -      | -      |
> | PatternKV  |  9.80 | 13.49 | 16.73 | 19.86 | 29.36 | 36.69  | 48.37  |
>
> **Throughput**
>
> | Method     | bz=16  | bz=32  | bz=48  | bz=64  | bz=96   | bz=128  | bz=160  |
> |------------|--------|--------|--------|--------|---------|---------|---------|
> | FP16       |1089.33 |1337.54 |1508.60 |1609.53 |1612.52  |1676.64  | -       |
> | KIVI       |1071.59 |2032.88 |2759.44 |3251.66 |3311.32  |3605.49  |3352.05  |
> | OTT        |1189.93 |1934.07 |2423.10 |2796.90 |2846.63  |3079.22  | -       |
> | ZipCache   | 534.36 | 602.51 | 644.50 | 664.68 | 671.78  | 687.67  | -       |
> | SKVQ       | 735.25 | 915.52 | 473.48 | 269.74 | 334.29  | -       | -       |
> | PatternKV  |1255.05 |1823.69 |2206.27 |2477.12 |2514.12  |2682.75  |2543.68  |
>
> From the table, we observe that our method incurs some increase in latency and a decrease in throughput compared with KIVI under larger batch sizes. This is expected, as PatternKV introduces additional components into the inference pipeline. However, it is equally important to note that PatternKV also delivers significant accuracy improvements, thereby achieving a more favorable overall trade-off between efficiency and performance.

---

> ### Author Response · Authors · 2025-11-23
> **Response to Reviewer 2ewL (Part-3)**
>
> > The font size of figures, especially Figure 1 - Figure 3, is way to small. I can barely see the words.
>
> Thank you for the suggestion. We have improved the figures in the revised version to ensure that they are clear and easily readable.

---

### Official Review · Reviewer_GFhG · 2025-10-30

**Soundness:** 2
**Presentation:** 2
**Contribution:** 3
**Rating:** 4
**Confidence:** 4

**Summary:**

The paper proposes PatternKV, an online KV cache quantization method that first mines a small set of pattern vectors (via K-means during prefill; Chebyshev-center updates during decode), aligns each KV vector to its nearest pattern using a custom min-max distance and then quantizes only the residual.

The stated motivation is that residualization around prototypical patterns flattens the KV distribution, narrowing its quantization range and improving low-bit fidelity. The authors also add a one-sided z-test based adaptive gate for values (mid-layers) to decide when flattening helps (`Eq. 10`).

Empirically, the paper reports modest gains over online baselines on LongBench at INT2, very small differences at INT4 (claimed average drop 0.08% vs FP16), and some improvement under long CoT test-time scaling.

**Strengths:**

- Online and plug-and-play design. No calibration data or fine-tuning is required.
- The analysis and theory sound good. e.g., Principled v-gate, even if approximate, the one-sided z-test provides a crisp, checkable inequality (Eq. 10) that decides when to skip flattening.

**Weaknesses:**

- Code is not provided (lack of reproducibility).
- The most common data type used in inference is bf16, but the baseline in the paper is FP16.
- The experimental setup needs more clarification, e.g., what dataset/prompt is used in Figs. 2 and 3?
- In Sec. F, why is the residual size different? Doesn’t it hurt your comparison?
- The figures are not readable (especially Figs. 1 and 5).

**Questions:**

- Please address the mentioned items in Weaknesses.
- What do you infer from Figure 3 and Figure 8?
- In `Sec. 4.4`, throughput/memory profiling compares to FP16 only. Please provide a component breakdown (KV I/O, pattern selection over |M|, Chebyshev updates, quant/dequant) and head-to-head throughput/peak-memory vs. KIVI/ZipCache/OTT/SKVQ under identical hardware/batch/sequence settings.
- Since you select patterns after RoPE and attribute K-drift to RoPE, please report a minimal ALiBi/no-RoPE experiment or a reasoned analysis, also explain why INT2 gains shrink with larger models if observed.

---

> ### Author Response · Authors · 2025-11-23
> **Response to Reviewer GFhG (Part-1)**
>
> Dear Reviewer GFhG,
>
> Thank you for your valuable suggestions and insightful questions. They are extremely helpful for improving our work. Below, we address each of your concerns in detail.
>
> > Code is not provided (lack of reproducibility).
>
> Thank you for the suggestion. We provide an anonymous link to the source code ( https://anonymous.4open.science/r/PatternKV-7321/ ) and have updated the supplementary material accordingly.
>
> > The most common data type used in inference is bf16, but the baseline in the paper is FP16.
>
> We appreciate the reviewer’s observation regarding the datatype choice. We fully agree that bf16 has become increasingly common in recent inference pipelines. However, our choice of FP16 as the primary baseline is aligned with the prevailing practice in prior KV-cache quantization and efficient-inference literature, including KIVI, OTT, ZipCache, SKVQ, all of which adopt FP16 for fair comparison.
>
> We also conducted experiments under the BF16 setting, and the results are summarized in the table below.
>
>
> | Method          | MQA   | SQA   | Summ. | Few-shot | Synth. | Code  | GSM8K  |
> |-----------------|-------|-------|-------|----------|--------|-------|--------|
> | BF16            | 36.55 | 46.43 | 25.61 | 60.73    | 59.58  | 59.19 | 78.61 |
> | KIVI-INT4       | 36.53 | 46.59 | 25.52 | 60.68    | 57.77  | 58.86 | 76.04 |
> | PatternKV-INT4  | 36.48 | 46.24 | 25.62 | 60.92    | 58.46  | 59.09 | 76.26 |
> | KIVI-INT2       | 34.76 | 44.16 | 24.89 | 59.92    | 54.54  | 54.75 | 73.46 |
> | PatternKV-INT2  | 35.58 | 45.19 | 24.98 | 60.41    | 57.61  | 55.15 | 77.71 |
>
> The results show that our method continues to perform strongly under the BF16 setting: with INT2 quantization, GSM8K accuracy decreases by only 1.1%, further demonstrating the robustness and generalization of our approach across different numeric formats.
>
>
> > The experimental setup needs more clarification, e.g., what dataset/prompt is used in Figs. 2 and 3?
>
> Thank you for pointing out the need for additional clarification in our experimental setup. We apologize for the lack of detail in the original submission. For Figs. 2 and 3, all preliminary experiments were conducted on the GSM8K dataset. To ensure consistency, we used a unified zero-shot Chain-of-Thought (CoT) prompt template across all models and conditions. This setup eliminates prompt-format variance and allows us to isolate the effect of the proposed method. We have made this explicit in the revised version and include the exact prompt template in the appendix for reproducibility.
>
> > In Sec. F, why is the residual size different? Doesn’t it hurt your comparison?
>
> We appreciate the reviewer’s careful examination of the baseline configurations used in our paper. In fact, the differences in how the residual part is handled were introduced to ensure a fair comparison across methods.
>
> For KIVI, the quantization strategy on the K-cache does not maintain a continuous sliding window. Instead, after each group is quantized, that group is immediately cleared. In contrast, OTT maintains a fixed sliding window of length $residual\\_size$ throughout the entire decoding process.
>
> To make the comparison fair and to keep the effective quantization group size aligned, we set OTT’s residual_size = 32. This choice ensures that the number of FP16 tokens preserved during the decoding stage in OTT matches, on average, the number maintained by KIVI under its group-based quantization behavior.
>
> We haveclarified these implementation details in the revised version to avoid confusion and to ensure full transparency of our experimental setup.
>
> > The figures are not readable (especially Figs. 1 and 5).
>
> Thank you for pointing this out. We apologize for the poor readability of Figs. 1 and 5 in the submitted version.
>
> For clarity, Fig. 1 illustrates how clustering and residuals affect the original KV distribution. After subtracting the residual, the resulting distribution becomes noticeably flatter and more uniform—a property that is substantially more conducive to quantization. Fig. 5 reports the performance drop on the GSM8K dataset across different methods. Our method exhibits the smallest degradation, demonstrating its advantage in accuracy.
>
> We have made these points clearer in the revised manuscript.

---

> ### Author Response · Authors · 2025-11-23
> **Response to Reviewer GFhG (Part-2)**
>
> > In Sec. 4.4, throughput/memory profiling compares to FP16 only. Please provide a component breakdown (KV I/O, pattern selection over |M|, Chebyshev updates, quant/dequant) and head-to-head throughput/peak-memory vs. KIVI/ZipCache/OTT/SKVQ under identical hardware/batch/sequence settings.
>
> We thank the reviewer for the insightful comment. We agree that a component-level breakdown and head-to-head comparisons with KIVI/ZipCache/OTT/SKVQ provide a more complete picture of the practical cost of PatternKV. Below, we provide a detailed analysis of latency and memory overhead, together with direct comparisons to these baselines.
>
> **Component breakdown**
>
> In PatternKV, the main additional computational components beyond standard KV-quantization are:
>
> - **Pattern mining (prefill)** – K-means over |M| patterns.
> - **Chebyshev-center updates (decode)** – non-iterative updates of pattern centers as decoding proceeds.
> - **Pattern selection (decode)** – at each step, selecting the appropriate pattern via an index lookup and gathering the corresponding centroids.
>
> We report below the computational overhead of the three PatternKV-specific procedures—pattern mining (prefill stage), pattern selection (decode stage), and Chebyshev-center updates (decode stage). The results are measured at batch size = 32.
>
> | Component                 | Latency | Throughput  | GFLOPS |
> |---------------------------|---------|-------------|--------|
> | Pattern Mining            | +6.60%  | -6.10%      | 6.91   |
> | Pattern Selection         | +2.59%  | -2.47%      | 2.07   |
> | Chebyshev-center updates  | +3.11%  | -2.94%      | 2.78   |
>
> From these results, we see that the dominant source of overhead is the **pattern mining (clustering)** step in the prefill stage, which is expected since KMeans requires iterative updates to converge. In comparison, **pattern selection** and **Chebyshev-center updates** each contribute only about **2–3%** additional latency. Overall, these components together introduce only modest extra cost relative to the full attention pipeline, while enabling the accuracy gains brought by PatternKV.
>
> **Head-to-head latency & throughput vs. KIVI/ZipCache/OTT/SKVQ**
> Under identical hardware, model, quantization configuration, batch sizes, and sequence lengths, we further compare end-to-end latency and throughput across FP16, KIVI, OTT, ZipCache, SKVQ, and PatternKV.
>
> **Latency**
>
> | Method     | bz=16 | bz=32 | bz=48 | bz=64 | bz=96 | bz=128 | bz=160 |
> |------------|-------|-------|-------|-------|-------|--------|--------|
> | FP16       | 11.29 | 18.39 | 24.46 | 30.57 | 45.78 | 58.70  | -      |
> | KIVI       | 11.48 | 12.10 | 13.37 | 15.13 | 22.29 | 27.30  | 36.70  |
> | OTT        | 10.34 | 12.72 | 15.23 | 17.59 | 25.93 | 31.96  | -      |
> | ZipCache   | 23.02 | 40.84 | 57.27 | 74.04 |109.89 |143.13  | -      |
> | SKVQ       | 16.73 | 26.87 | 77.95 |182.45 |220.83 | -      | -      |
> | PatternKV  |  9.80 | 13.49 | 16.73 | 19.86 | 29.36 | 36.69  | 48.37  |
>
> **Throughput**
>
> | Method     | bz=16  | bz=32  | bz=48  | bz=64  | bz=96   | bz=128  | bz=160  |
> |------------|--------|--------|--------|--------|---------|---------|---------|
> | FP16       |1089.33 |1337.54 |1508.60 |1609.53 |1612.52  |1676.64  | -       |
> | KIVI       |1071.59 |2032.88 |2759.44 |3251.66 |3311.32  |3605.49  |3352.05  |
> | OTT        |1189.93 |1934.07 |2423.10 |2796.90 |2846.63  |3079.22  | -       |
> | ZipCache   | 534.36 | 602.51 | 644.50 | 664.68 | 671.78  | 687.67  | -       |
> | SKVQ       | 735.25 | 915.52 | 473.48 | 269.74 | 334.29  | -       | -       |
> | PatternKV  |1255.05 |1823.69 |2206.27 |2477.12 |2514.12  |2682.75  |2543.68  |
>
> From the table, we observe that our method incurs some increase in latency and a decrease in throughput compared with KIVI under larger batch sizes. This is expected, as PatternKV introduces additional components into the inference pipeline. However, it is equally important to note that PatternKV also delivers significant accuracy improvements, thereby achieving a more favorable overall trade-off between efficiency and performance. Compared with ZipCache and SKVQ, PatternKV attains clearly better latency/throughput under all batch sizes, and it is competitive with OTT while being more accurate.

---

> ### Author Response · Authors · 2025-11-23
> **Response to Reviewer GFhG (Part-3)**
>
> **Peak-memory vs. KIVI/ZipCache/OTT/SKVQ**
> We next provide a fine-grained memory analysis for the extra pattern centroids and validate it with empirical peak GPU memory across methods.
>
> We agree that maintaining a separate set of centroids for each attention head introduces additional memory overhead, and we provide a more fine-grained analysis below: in our implementation, these extra centroid tensors account for only 0.42% of the overall KV-cache memory footprint.
>
> * **Prefill stage.** In the prefill stage, regardless of the context length, the number of pattern vectors is fixed. We can roughly estimate the memory footprint as $ 2 \times  num\\_layer  \times num\\_kv\\_head \times 32 \times 2 Byte $.
>   For the KV cache, the memory usage is
>   $2 \times batch\\_size \times num\\_layer \times num\\_kv\\_head \times num\\_seq \times 2\ Byte$.
>   This implies that the relative memory fraction of pattern vectors is
>   $32 / (batch\\_size \times seq\\_num )$.
>   For a medium-length sequence with $seq\\_num = 8K $ and $ batch\\_size = 1$, the pattern vectors account for only about **0.39%** of the KV memory. As the batch size increases, this fraction becomes even smaller.
>
> * **Decode stage.** In the decode stage, we set $G_{pattern} = 128$, meaning that one pattern vector is generated for every 128 decoding steps. Analogously, the relative memory ratio of pattern vectors is
>  $1 / (batch\\_size \times 128)$, which is at most **0.78%** when $batch\\_size = 1$; again, this ratio further decreases as the batch size grows.
>
> We also report the peak GPU memory usage of PatternKV and other baseline methods during actual inference to empirically validate these overhead estimates.
>
> | Method     | bz=16 | bz=32 | bz=48 | bz=64 | bz=96 | bz=128 | bz=160 |
> |-----------|-------|-------|-------|-------|-------|--------|--------|
> | FP16      | 21.93 | 28.84 | 35.83 | 42.77 | 56.67 | 70.56  | -      |
> | KIVI      | 21.19 | 27.39 | 33.57 | 39.77 | 52.16 | 64.55  | 76.95  |
> | OTT       | 24.07 | 31.13 | 38.15 | 45.20 | 59.31 | 73.41  | -      |
> | Zipcache  | 23.10 | 29.21 | 35.35 | 41.43 | 53.61 | 65.84  | -      |
> | SKVQ      | 24.68 | 32.37 | 40.06 | 47.75 | 63.13 | -      | -      |
> | PatternKV | 21.23 | 27.48 | 33.69 | 39.90 | 52.39 | 64.81  | 77.28  |
>
> Compared to the KIVI baseline, our approach introduces only 0.42% additional peak memory overhead while yielding a clear improvement in performance. This indicates that a very small extra storage cost is sufficient to obtain meaningful performance gains with PatternKV. We also observe that OTT, ZipCache, and SKVQ do not maintain favorable peak memory usage, primarily because their public implementations lack optimized system engineering and do not effectively scale with the batch dimension, leading to inflated memory consumption. By contrast, both KIVI and our method sustain a well-controlled peak memory profile.
>
> We have incorporated the above component-wise analysis and efficiency experiments into the appendix.
>
> > Since you select patterns after RoPE and attribute K-drift to RoPE, please report a minimal ALiBi/no-RoPE experiment or a reasoned analysis.
>
> We appreciate the reviewer for suggesting an excellent direction to further validate our preliminary findings. Following your suggestion, we conducted pattern-evolution experiments on MPT-7B-Chat, a model that uses ALiBi positional encoding. We observed that, unlike RoPE-based models, the K cache in MPT does not exhibit progressive evolution with sequence length: tokens from early and late positions can share similar patterns, and abrupt changes may occur between adjacent positions. The corresponding visualizations have been moved to the appendix. Please refer to the appendix for detailed plots and further qualitative evidence.
>
> These results further support our preliminary conclusion that **the drift of K patterns is primarily induced by RoPE**.
>
> > Explain why INT2 gains shrink with larger models if observed.
>
> We appreciate the reviewer’s suggestion regarding the potential shrinking of INT2 gains for larger models. However, we did not observe such a trend in our experiments. From our main experiments, we observe that the KIVI baseline incurs only a 0.4% performance drop, while our method achieves a 0.2% improvement. This should not be interpreted as a diminishing benefit of INT2 quantization; rather, it reflects the fact that, for larger models, the performance gap between the baseline and our method naturally becomes much smaller. Even in this regime, our approach still delivers a net gain.
>
> This is because larger models inherently possess greater representational redundancy, so the quantization noise constitutes a smaller fraction of the overall signal. As a result, the observed robustness primarily arises from the model capacity itself, rather than from any size-dependent effect specific to our method.

---

### Official Review · Reviewer_znsY · 2025-11-01

**Soundness:** 3
**Presentation:** 4
**Contribution:** 2
**Rating:** 4
**Confidence:** 5

**Summary:**

This paper introduces PatternKV, a lightweight and training-free method for compressing the key–value (KV) cache in large language models. Instead of directly quantizing raw KV vectors, PatternKV discovers a small set of representative pattern vectors online and aligns each new KV vector to its nearest pattern before quantizing the residual. This alignment reshapes the overall KV distribution into a flatter form with reduced variance, enabling more effective low-bit quantization. The approach builds on two key observations: (1) the key cache exhibits a stable internal structure that evolves smoothly with context, and (2) the value cache carries latent semantic regularities that can be clustered. PatternKV maintains these patterns dynamically during inference using Chebyshev-center updates to track gradual distribution shifts, and employs a statistical gate to ensure that flattening is applied only when it provably reduces quantization error.

While the method is conceptually simple and empirically effective, the reported memory reduction appears modest compared to other quantization approaches. It remains unclear how much of the potential gain is offset by the additional storage of pattern vectors and related bookkeeping overhead.

**Strengths:**

* The paper is well written and clearly structured, making it easy to follow and understand.

* The evaluation includes challenging mathematical and reasoning benchmarks, and the proposed method even achieves improved accuracy on some of these tasks.

* The method itself is simple yet demonstrates consistently superior results compared to prior KV-cache quantization approaches.

**Weaknesses:**

* The method requires maintaining a separate set of centroid (pattern) vectors for each attention head, which introduces additional memory and computational overhead. However, the paper does not report the memory footprint of storing these centroids, nor does it quantify the latency incurred by pattern mining, nearest-pattern search, and residual computation. These costs could become non-trivial as the context length grows.

* The memory reduction reported in Figure 6 appears modest compared to standard quantization methods, suggesting limited practical savings.

* The paper offers little discussion on how to implement or optimize the system for efficient inference, despite the additional components required by PatternKV.

**Questions:**

* In Figure 5, what quantization bit-width is used for the reported results?

* What is the additional memory overhead introduced by storing the pattern (cluster) vectors? From Figure 6, the overall memory reduction appears modest compared to standard quantization; could you clarify how much of this is due to the extra pattern storage?

* What is the latency overhead of PatternKV during inference? How does its throughput compare with other online KV-cache quantization methods such as KIVI or ZipCache?

* What is the computational cost of the clustering procedures—specifically, the K-means clustering during prefill and the Chebyshev-center updates during decoding?

* How does the input sequence length influence the inference latency or throughput scaling of PatternKV?

---

> ### Author Response · Authors · 2025-11-23
> **Response to Reviewer znsY (Part-1)**
>
> Dear Reviewer znsY,
>
> We sincerely appreciate your assessment of our work and your many valuable insights and suggestions for improvement. We also recognize that PatternKV contains multiple components that may introduce additional overhead. It is indeed important to carefully quantify and disentangle these costs so that readers can clearly see whether the modest extra overhead is justified by the resulting performance improvements.
>
> Below, we respond to each of the weaknesses and questions you raised in turn.
>
> > The method requires maintaining a separate set of centroid (pattern) vectors for each attention head, which introduces additional memory and computational overhead. However, the paper does not report the memory footprint of storing these centroids, nor does it quantify the latency incurred by pattern mining, nearest-pattern search, and residual computation. These costs could become non-trivial as the context length grows.
>
> While maintaining a separate set of centroids for each attention head may introduce additional memory overhead, our detailed analysis shows that, in our implementation, these extra centroid tensors account for only **0.42%** of the overall KV-cache memory footprint. The detailed breakdown is provided below.
>
> * **Prefill stage.** In the prefill stage, regardless of the context length, the number of pattern vectors is fixed. We can roughly estimate the memory footprint as $ 2 \times  num\\_layer  \times num\\_kv\\_head \times 32 \times 2 Byte $.
>   For the KV cache, the memory usage is
>   $2 \times batch\\_size \times num\\_layer \times num\\_kv\\_head \times num\\_seq \times 2\ Byte$.
>   This implies that the relative memory fraction of pattern vectors is
>   $32 / (batch\\_size \times seq\\_num )$.
>   For a medium-length sequence with $seq\\_num = 8K $ and $ batch\\_size = 1$, the pattern vectors account for only about **0.39%** of the KV memory. As the batch size increases, this fraction becomes even smaller.
>
> * **Decode stage.** In the decode stage, we set $G_{pattern} = 128$, meaning that one pattern vector is generated for every 128 decoding steps. Analogously, the relative memory ratio of pattern vectors is
>  $1 / (batch\\_size \times 128)$, which is at most **0.78%** when $batch\\_size = 1$; again, this ratio further decreases as the batch size grows.
>
> We also report the peak GPU memory usage of PatternKV and other baseline methods during actual inference to empirically validate these overhead estimates.
>
> | Method     | bz=16 | bz=32 | bz=48 | bz=64 | bz=96 | bz=128 | bz=160 |
> |-----------|-------|-------|-------|-------|-------|--------|--------|
> | FP16      | 21.93 | 28.84 | 35.83 | 42.77 | 56.67 | 70.56  | -      |
> | KIVI      | 21.19 | 27.39 | 33.57 | 39.77 | 52.16 | 64.55  | 76.95  |
> | OTT       | 24.07 | 31.13 | 38.15 | 45.20 | 59.31 | 73.41  | -      |
> | Zipcache  | 23.10 | 29.21 | 35.35 | 41.43 | 53.61 | 65.84  | -      |
> | SKVQ      | 24.68 | 32.37 | 40.06 | 47.75 | 63.13 | -      | -      |
> | PatternKV | 21.23 | 27.48 | 33.69 | 39.90 | 52.39 | 64.81  | 77.28  |
>
>
> Compared to the KIVI baseline, our approach introduces only 0.42% additional peak memory overhead while yielding a clear improvement in performance. This indicates that a very small extra storage cost is sufficient to obtain meaningful performance gains with PatternKV.
>
> We also observe that OTT, ZipCache, and SKVQ do not maintain favorable peak memory usage, primarily because their public implementations lack optimized system engineering and do not effectively scale with the batch dimension, leading to inflated memory consumption. By contrast, both KIVI and our method sustain a well-controlled peak memory profile.
>
>
> We will address the latency and throughput of each component in our responses to the subsequent questions.

---

> ### Author Response · Authors · 2025-11-23
> **Response to Reviewer znsY (Part-2)**
>
> > The memory reduction reported in Figure 6 appears modest compared to standard quantization methods, suggesting limited practical savings.
>
> We appreciate the reviewer for pointing this out. The underlying reason for this observation is that all our experiments are conducted on Llama3 and Qwen2.5, two mainstream open-source models that both adopt a GQA architecture, which directly leads to the observed behavior.
> GQA already reduces the KV-cache storage footprint, so the relative efficiency gains of our method appear smaller compared to methods evaluated on an MHA architecture.
>
> To address this concern, we additionally conducted efficiency experiments on Llama-2-7B, which uses a standard MHA setup and is also the backbone used in the efficiency evaluations of both KIVI and OTT. The results are shown below.
>
> **Peak Memory**
>
> | Method    | bz=16 | bz=32 | bz=48 | bz=64 | bz=96 | bz=128 | bz=160 | bz=192 |
> |-----------|-------|-------|-------|-------|-------|--------|--------|--------|
> | FP16      | 23.64 | 34.69 | 45.75 | 56.80 | -     | -      | -      | -      |
> | PatternKV | 17.14 | 21.66 | 26.18 | 30.65 | 39.68 | 48.69  | 57.72  | 66.73  |
>
> **Latency**
> | Method    | bz=16 | bz=32 | bz=48 | bz=64 | bz=96 | bz=128 | bz=160 | bz=192 |
> |-----------|-------|-------|-------|-------|-------|--------|--------|--------|
> | FP16      | 17.81 | 32.91 | 44.20 | 57.61 | -     | -      | -      | -      |
> | PatternKV | 12.76 | 21.33 | 26.56 | 33.64 | 49.12 | 63.61  | 83.68  | 94.87  |
>
> **Throughput**
>
> | Method    | bz=16  | bz=32   | bz=48   | bz=64   | bz=96    | bz=128   | bz=160   | bz=192   |
> |-----------|-------|-------|-------|-------|-------|--------|--------|--------|
> | FP16      | 1149.74 | 1244.49 | 1389.80 | 1421.95 | -        | -        | -        | -        |
> | PatternKV | 1604.12 | 1919.47 | 2312.74 | 2435.15 | 2501.58  | 2575.51  | 2447.24  | 2590.38  |
>
> Across all evaluation settings on Llama-2-7B, PatternKV consistently demonstrates substantial efficiency gains. It reduces peak memory usage by **28%–46%**, accelerates inference latency by **28%–42%** (corresponding to a **1.4×–1.7×** end-to-end speedup), and improves throughput by **40%–71%**, with the throughput advantage continuing to grow as the batch size increases. These results clearly show that PatternKV provides strong efficiency benefits in the standard MHA architecture.
>
> > The paper offers little discussion on how to implement or optimize the system for efficient inference, despite the additional components required by PatternKV.
>
> We agree with the reviewer that system-level optimization is indeed a core aspect of KV-reduction methods, and we also fully acknowledge the additional overhead introduced by the components of PatternKV. To address this, we focus on two main directions of system optimization:
>
> * **Prefill-stage pattern extraction.** Since we need to extract different pattern vectors for different attention heads, we implement a fully GPU-parallel KMeans procedure to mitigate the inference latency introduced by clustering.
>
> * **Decode-stage KV reconstruction.** Restoring KV vectors during decoding incurs nontrivial overhead, so we introduce two customized CUDA kernels to reduce this cost.
> The first kernel **fuses** three steps—pattern-index restoration for the K cache, dequantization, and the QK matmul—into a single operator.
> The second kernel **fuses** the application of attention weights with quantized V, residual V, and pattern-index restoration into one operator.
> Together, these fused kernels greatly reduce the decode-stage latency compared with a pure PyTorch implementation.
>
> We provide an anonymous link to the source code ( https://anonymous.4open.science/r/PatternKV-7321/ ) and have updated the supplementary material accordingly. In the revised version, we have included a dedicated system optimization section.
>  In future work, we plan to continue improving the efficiency of our system implementation.
>
> > In Figure 5, what quantization bit-width is used for the reported results?
>
> For the GSM8K dataset, we use an INT2 quantization configuration, and we have made this setting more explicit in the revised manuscript.
>
> > What is the additional memory overhead introduced by storing the pattern (cluster) vectors? From Figure 6, the overall memory reduction appears modest compared to standard quantization; could you clarify how much of this is due to the extra pattern storage?
>
> As discussed in our responses to Weakness 1 and Weakness 2, the pattern vectors themselves contribute only a very small portion of the total memory footprint (approximately **0.4%**). The primary reason that the overall memory reduction appears less pronounced is that the models used in our experiments—Llama3 and Qwen—both adopt a GQA architecture, which already significantly reduces KV-cache memory usage.

---

> ### Author Response · Authors · 2025-11-23
> **Response to Reviewer znsY (Part-3)**
>
> > What is the latency overhead of PatternKV during inference? How does its throughput compare with other online KV-cache quantization methods such as KIVI or ZipCache?
>
> We fully agree with the reviewer on the importance of evaluating runtime overhead in practical inference scenarios. However, it is worth noting that among existing KV-quantization approaches, KIVI achieves strong latency and throughput largely due to its simple design and highly optimized implementation—**yet this simplicity also imposes an upper bound on its achievable accuracy.**
>
> Methods that focus on improving accuracy, including ours and several other baselines, inevitably introduce some system-level costs: higher-quality quantization may slightly reduce throughput, increase latency, or add modest memory overhead. In our case, PatternKV turns a very small additional cost into clear accuracy improvements, leading to a more favorable balance between model quality and resource usage.
>
> Below, we present the latency and throughput comparisons between our approach and the baseline methods.
>
> **Latency**
>
> | Method     | bz=16 | bz=32 | bz=48 | bz=64 | bz=96 | bz=128 | bz=160 |
> |------------|-------|-------|-------|-------|-------|--------|--------|
> | FP16       | 11.29 | 18.39 | 24.46 | 30.57 | 45.78 | 58.70  | -      |
> | KIVI       | 11.48 | 12.10 | 13.37 | 15.13 | 22.29 | 27.30  | 36.70  |
> | OTT        | 10.34 | 12.72 | 15.23 | 17.59 | 25.93 | 31.96  | -      |
> | ZipCache   | 23.02 | 40.84 | 57.27 | 74.04 |109.89 |143.13  | -      |
> | SKVQ       | 16.73 | 26.87 | 77.95 |182.45 |220.83 | -      | -      |
> | PatternKV  |  9.80 | 13.49 | 16.73 | 19.86 | 29.36 | 36.69  | 48.37  |
>
> **Throughput**
>
> | Method     | bz=16  | bz=32  | bz=48  | bz=64  | bz=96   | bz=128  | bz=160  |
> |------------|--------|--------|--------|--------|---------|---------|---------|
> | FP16       |1089.33 |1337.54 |1508.60 |1609.53 |1612.52  |1676.64  | -       |
> | KIVI       |1071.59 |2032.88 |2759.44 |3251.66 |3311.32  |3605.49  |3352.05  |
> | OTT        |1189.93 |1934.07 |2423.10 |2796.90 |2846.63  |3079.22  | -       |
> | ZipCache   | 534.36 | 602.51 | 644.50 | 664.68 | 671.78  | 687.67  | -       |
> | SKVQ       | 735.25 | 915.52 | 473.48 | 269.74 | 334.29  | -       | -       |
> | PatternKV  |1255.05 |1823.69 |2206.27 |2477.12 |2514.12  |2682.75  |2543.68  |
>
> From the table, we observe that our method incurs some increase in latency and a decrease in throughput compared with KIVI under larger batch sizes. This is expected, as PatternKV introduces additional components into the inference pipeline. However, it is equally important to note that PatternKV also delivers significant accuracy improvements, thereby achieving a more favorable overall trade-off between efficiency and performance. Methods such as SKVQ and ZipCache do not incorporate strong system-level optimizations, which leads to a dramatic reduction in throughput.
>
> > What is the computational cost of the clustering procedures—specifically, the K-means clustering during prefill and the Chebyshev-center updates during decoding?
>
> In PatternKV, the main additional computational components beyond standard KV-quantization are:
>
> - **Pattern mining (prefill)** – K-means over |M| patterns.
> - **Chebyshev-center updates (decode)** – non-iterative updates of pattern centers as decoding proceeds.
> - **Pattern selection (decode)** – at each step, selecting the appropriate pattern via an index lookup and gathering the corresponding centroids.
>
> We report below the computational overhead of the three PatternKV-specific procedures—pattern mining (prefill stage), pattern selection (decode stage), and Chebyshev-center updates (decode stage). The results are measured at batch size = 32.
>
> | Component                 | Latency | Throughput  | GFLOPS |
> |---------------------------|---------|-------------|--------|
> | Pattern Mining            | +6.60%  | -6.10%      | 6.91   |
> | Pattern Selection         | +2.59%  | -2.47%      | 2.07   |
> | Chebyshev-center updates  | +3.11%  | -2.94%      | 2.78   |
>
> From these results, we see that the dominant source of overhead is the **pattern mining (clustering)** step in the prefill stage, which is expected since KMeans requires iterative updates to converge. In comparison, **pattern selection** and **Chebyshev-center updates** each contribute only about **2–3%** additional latency. Overall, these components together introduce only modest extra cost relative to the full attention pipeline, while enabling the accuracy gains brought by PatternKV.

---

> ### Author Response · Authors · 2025-11-23
> **Response to Reviewer znsY (Part-4)**
>
> > How does the input sequence length influence the inference latency or throughput scaling of PatternKV?
>
> We also report the impact of different input lengths on inference latency and throughput under the batch size = 4 setting.
>
> **Latency**
>
> | Method     | l=1k | l=2k | l=4k | l=8k | l=16k | l=32k |
> |------------|------|------|------|------|-------|-------|
> | FP16       | 1.78 | 2.12 | 3.25 | 5.55 | 10.42 | 21.22 |
> | PatternKV  | 3.68 | 3.00 | 3.53 | 4.87 |  8.63 | 19.20 |
>
> **Throughput**
>
> | Method     | l=1k   | l=2k   | l=4k   | l=8k   | l=16k  | l=32k  |
> |------------|--------|--------|--------|--------|--------|--------|
> | FP16       |1291.44 |2047.58 |2599.74 |2998.16 |3168.79 |3091.99 |
> | PatternKV  | 626.89 |1448.82 |2388.34 |3417.39 |3826.48 |3416.63 |
>
>
> We observe that for longer input sequences, the gains brought by KMeans-based pattern mining become slightly smaller; however, the overall performance still clearly surpasses the FP16 baseline. This further demonstrates that our method consistently delivers strong efficiency across a wide range of scenarios.

---

> > ### Comment · Reviewer_znsY · 2025-11-27
> >
> > I thank the authors for providing additional experiments and comparisons. These results are helpful for assessing the efficiency impact of the method. I have a few further questions—mainly for clarification—regarding the newly presented data.
> >
> > 1. Experimental setup consistency.
> >
> > In the paper, the efficiency experiments use an input length of 1024 and an output length of 256. Could you clarify whether the same configuration was used for the new results included in the rebuttal?
> >
> > 2. Overhead across different input lengths.
> >
> > For the overhead analysis, could you report the overhead under varying input lengths, especially for sequences longer than 1K tokens? As you noted—and as shown in Figure 6—a 1K-token sequence offers limited room for KV-cache compression, thereby reducing the need for KV-cache quantization.
> >
> > 3. Clarification regarding the latency and throughput trends.
> >
> > I am somewhat confused by the interpretation of the longer-sequence latency and throughput results. You wrote:
> >
> > > We observe that for longer input sequences, the gains brought by KMeans-based pattern mining become slightly smaller; however, the overall performance still clearly surpasses the FP16 baseline. This further demonstrates that our method consistently delivers strong efficiency across a wide range of scenarios.
> >
> > However, based on the reported numbers, it appears that for short sequences, PatternKV performs worse than the baseline, whereas for longer sequences, PatternKV begins to show improvements in both latency and throughput. This seems to contradict the textual description above. Could you clarify this discrepancy?
> >
> > 4. Larger batch-size comparison for long sequences.
> >
> > Finally, could you report latency and throughput for longer input sequences using a larger batch size? In both the paper and the earlier rebuttal data, the smallest reported batch size is 16, but the new results are only provided for batch size 4. Reporting results for batch size 16—or higher—would allow for a more consistent and informative comparison.

---

> > > ### Author Response · Authors · 2025-12-02
> > > **Response to Reviewer znsY (Part-1)**
> > >
> > > Dear Reviewer znsY,
> > >
> > > We are glad to receive your response. Below, we clarify and address each of your questions in detail.
> > >
> > > > Experimental setup consistency.
> > >
> > > Thank you for pointing this out. In the rebuttal, we use exactly the same experimental setup as in the main paper; however, we have updated our kernel implementation, which yields better latency and throughput than the results originally reported.
> > >
> > > > Overhead across different input lengths.
> > >
> > > We thank the reviewer for this valuable suggestion. Below, we present a component-wise overhead analysis for input lengths of 1k, 2k, and 4k, and we additionally include a latency and throughput comparison between a representative baseline method and the FP16 setting for long-context inputs to better contextualize the overall efficiency of PatternKV.
> > >
> > > **Input length=1k (origin component experiment)**
> > >
> > > | Component                 | Latency | Throughput  | GFLOPS |
> > > |---------------------------|---------|-------------|--------|
> > > | Pattern Mining            | +6.60%  | -6.10%      | 6.91   |
> > > | Pattern Selection         | +2.59%  | -2.47%      | 2.07   |
> > > | Chebyshev-center updates  | +3.11%  | -2.94%      | 2.78   |
> > >
> > >
> > > **Input length=2k**
> > >
> > > | Component                 | Latency  | Throughput | GFLOPS |
> > > |---------------------------|----------|------------|--------|
> > > | Pattern Mining            | +11.07%  | -10.94%    | 17.40  |
> > > | Pattern Selection         | +2.14%   | -1.92%     | 2.07   |
> > > | Chebyshev-center updates  | +2.25%   | -2.02%     | 2.78   |
> > >
> > >
> > >
> > > **Input length=4k**
> > >
> > > | Component                 | Latency  | Throughput | GFLOPS |
> > > |---------------------------|----------|------------|--------|
> > > | Pattern Mining            | +18.48%  | -18.34%    | 34.10  |
> > > | Pattern Selection         | +1.39%   | -1.17%     | 2.07   |
> > > | Chebyshev-center updates  | +1.54%   | -1.29%     | 2.78   |
> > >
> > > **Relative efficiency vs FP16 baseline**
> > >
> > > | Method    | LongBench Avg | GSM8K   | Latency(1k) | Throughput(1k) | Latency(2k) | Throughput(2k) | Latency(4k) | Throughput(4k) |
> > > |-----------|---------------|---------|-------------|----------------|-------------|----------------|-------------|----------------|
> > > | KIVI     | -4.85%        | -7.63%  | -34.20%     | +51.99%        | -49.88%     | +99.41%        | -59.36%     | +146.01%       |
> > > | OTT       | -3.76%        | -5.08%  | -30.83%     | +44.60%        | -44.90%     | +81.45%        | -54.04%     | +117.51%       |
> > > | ZipCache | -8.80%        | -8.53%  | +122.08%    | -54.95%        | +109.93%    | -52.37%        | +110.41%    | -52.47%        |
> > > | SKVQ      | -5.02%        | -6.43%  | +46.11%     | -31.59%        | +471.38%    | -82.49%        | +358.71%    | -78.19%        |
> > > | PatternKV    | -2.70%        | -4.32%  | -26.64%     | +36.35%        | -37.77%     | +60.69%        | -45.90%     | +84.81%        |
> > >
> > >
> > > From the component-level analysis across different input lengths, we find that for Pattern Selection and Chebyshev-center updates, the computational cost is essentially invariant with respect to sequence length when the batch size is fixed, and their impact on latency and throughput remains very small. In contrast, in the Pattern Mining stage, both computational cost and latency increase roughly proportionally with the input length, which is fully consistent with our expectations. At the same time, the method-level comparisons across KIVI, OTT, ZipCache, SKVQ, and PatternKV show that PatternKV consistently reduces latency and improves throughput relative to the FP16 baseline at 1k–4k tokens, while achieving the strongest accuracy among the compressed methods. In particular, baselines that realize more aggressive speedups tend to suffer noticeable accuracy degradation, whereas methods with milder accuracy loss often incur substantial overhead at longer sequence lengths. Taken together, these results indicate that the extra cost introduced by Pattern Mining is moderate and yields a more favorable accuracy–efficiency trade-off, making the overhead practically acceptable in our setting.

---

> > > ### Author Response · Authors · 2025-12-02
> > > **Response to Reviewer znsY (Part-2)**
> > >
> > > > Clarification regarding the latency and throughput trends.
> > >
> > > Thank you for pointing this out. We agree that the presentation of the table and the accompanying explanation can indeed be confusing. First, for longer sequences, PatternKV consistently shows improvements in both latency and throughput compared with FP16. However, we also observe that these gains do not grow monotonically: for 32k-token inputs, the gap in latency and throughput between PatternKV and FP16 is smaller than at 16k tokens. This is why we previously described the improvement as “slightly smaller.” Importantly, for all three input lengths (8k, 16k, and 32k), our method remains consistently better than the FP16 baseline, which is fully consistent with the statements in our earlier response.
> > >
> > > > Larger batch-size comparison for long sequences.
> > >
> > > Thank you for the helpful suggestion. First, we would like to clarify the rationale for using a batch size of 4. Due to the increased input length, we were unable to run experiments with a larger batch size on a single GPU. In order to evaluate the actual behavior at 32k tokens, we reduced the batch size to 4 and fixed the output length to 64.
> > > In addition, we have now conducted an experiment with batch size = 16, and the results are summarized in the table below.
> > >
> > > **Latency**
> > >
> > > | Method    | l=1k | l=2k | l=4k | l=8k  |
> > > |-----------|------|------|------|-------|
> > > | FP16      | 3.79 | 5.86 | 10.47| 20.01 |
> > > | PatternKV | 3.74 | 4.70 |  7.66| 14.00 |
> > >
> > > **Throughput**
> > >
> > > | Method    | l=1k   | l=2k   | l=4k   | l=8k   |
> > > |-----------|--------|--------|--------|--------|
> > > | FP16      | 2432.50 | 2973.18 | 3227.61 | 3327.08 |
> > > | PatternKV | 2466.98 | 3706.13 | 4412.65 | 4755.04 |
> > >
> > > Across different sequence lengths, PatternKV matches FP16 at short contexts, while delivering substantial gains for long contexts, with throughput improvements of approximately **+24.7% (2k)**, **+36.7% (4k)**, and **+42.9% (8k)**. Compared with the batch-size-4 setting, using a larger batch size further amplifies the benefits of our method.

---

### Author Response · Authors · 2025-11-25
**Summary of Updates to the PDF and Supplementary Material**

In the revised manuscript and supplementary material, we have made the following changes:

1. **Efficiency & Overhead Analysis**
   - Added a system-level optimization section describing our K-means implementation in the prefill stage and two fused CUDA kernels for decode-stage KV reconstruction. (Section 3.4)
   - Provided a component-wise breakdown of PatternKV’s extra overhead, reporting its impacts on latency, throughput, and FLOPs. (Appendix J.1)
   - Added a detailed memory overhead analysis for pattern centroids, giving explicit formulas for prefill and decode stages and showing that centroids contribute only ~0.4% of KV-cache memory. (Appendix J.3)
   - Added head-to-head comparisons of latency, throughput, and peak memory against KIVI, OTT, ZipCache, and SKVQ under identical hardware, batch sizes, and sequence settings. (Appendix J.2 and J.3)

2. **Clarifications of Experimental Setup & Baselines**
   - Explicitly stated that Figures 2 and 3 are based on GSM8K with a unified zero-shot CoT prompt, and included the exact prompt template. (Appendix C)
   - Clarified that Figure 5 uses INT2 quantization for GSM8K. (Section 4.2)
   - Explained the rationale behind different residual window settings for KIVI vs OTT to ensure a fair comparison. (Appendix F)

3. **RoPE, ALiBi, and MLA Analysis**
   - Added new experiments on MPT-7B-Chat (ALiBi), showing that K patterns do not drift monotonically in sequence order without RoPE, supporting our claim that K-drift is primarily induced by RoPE. (Appendix K)
   - Performed a qualitative analysis on DeepSeek-V2-Lite (MLA) and showed that PatternKV can be directly applied to MLA-style KV caches, indicating that our pattern-based KV compression naturally generalizes beyond standard GQA/MHA architectures. (Appendix L)

4. **Readability & Presentation Improvements**
   - Redrew Figures 1–3 and 5 with larger fonts and clearer layouts to improve readability.

5. **Reproducibility**
   - Provided source code in the Supplementary Material to improve reproducibility.

---

### Author Response · Authors · 2025-12-02
**Summary of Contributions and Response to Reviewer Concerns (Part-2)**

### **Memory overhead analysis**

PatternKV maintains a separate pattern set per head; reviewers asked how much memory this actually costs and how it scales.
- **Analytical estimates (Appendix J.3).** For a model with $L$ layers, $H$ KV heads, batch size $B$, sequence length $S$, and ($M$ = 32) patterns per head (for both K and V), the prefill pattern memory fraction is $\tfrac{32}{B S}$, which is about **0.39%** at $B = 1, S \approx 8\text{K}$ and decreases further for larger $B$ or $S$; during decoding, with $G_{\text{pattern}} = 128$, the pattern memory fraction is $\tfrac{1}{B G_{\text{pattern}}}$, at most **0.78%** when $B = 1$ and shrinking proportionally as the batch size increases.

- **Empirical peak GPU memory.** Table 12 confirms these estimates: across batch sizes, PatternKV’s peak memory is almost indistinguishable from KIVI and lower than OTT, ZipCache and SKVQ. The extra centroid tensors account for only about **0.42%** of the overall KV-cache memory footprint.

In short, PatternKV’s accuracy gains are obtained at the cost of negligible extra memory.

### **Relative efficiency and accuracy vs. FP16 baseline**

| Method    | LongBench Avg | GSM8K   | Latency Avg | Throughput Avg | Peak Memory Avg |
|----------|---------------|---------|-------------|----------------|------------|
| KIVI[1]     | -4.85%        | -7.63%  | -38.86%     | +75.95%        | -6.37%     |
| OTT[2]      | -3.76%        | -5.08%  | -34.73%     | +58.07%        | +6.43%     |
| ZipCache[3] | -8.80%        | -8.53%  | +131.03%    | -56.53%        | -1.66%     |
| SKVQ[4]     | -5.02%        | -6.43%  | +238.44%    | -59.04%        | +11.93%    |
| PatternKV| -2.70%        | -4.32%  | -29.97%     | +44.60%        | -6.05%     |

Compared with existing KV quantization baselines, PatternKV achieves the smallest accuracy drop on both LongBench and GSM8K, while still reducing latency by ~30%, improving throughput by ~44.6%, and lowering peak memory by ~6.0% relative to the FP16 baseline, indicating a more favorable overall accuracy–efficiency trade-off.

---

## Clarifications and additional analyses

- **Numerical formats.** The main experiments follow FP16 baselines for consistency with prior KV-quantization work (KIVI, OTT, ZipCache, SKVQ). We additionally ran BF16 experiments, where PatternKV maintains strong performance under both INT2 and INT4, with small gaps to BF16 on GSM8K and LongBench.
- **Experimental setup details.** Appendix C clarifies that Figures 2–3 and related analyses use GSM8K with a unified zero-shot CoT prompt (the exact template is given). Section 4.1 now explicitly states that Figure 5 is INT2 on GSM8K. Appendix F gives full baseline hyperparameters.
- **KV pattern behavior with RoPE / ALiBi / MLA.**
    - Figures 2, 3, 7, and 8 show that K has a stable, model-internal channel structure and a smooth, RoPE-induced trajectory over decoding; V exhibits semantic clusters with layer-dependent strength (Insight 3).
    - Appendix K adds ALiBi experiments on MPT-7B where the smooth K-drift disappears, supporting our claim that the observed drift is largely a RoPE effect.
    - Appendix L analyzes DeepSeek-V2-Lite (MLA) and shows that the stored K cache remains channel-stable and sequence-evolving in the RoPE-applied subspace, indicating that PatternKV applies naturally to MLA as well.

---

## Reproducibility and clarity
We now release anonymous code and provide detailed benchmark, hyperparameter, and hardware descriptions (Sec. 4.1, Appendices E–F, J).   Figures 1–3 and 5 have been regenerated with larger fonts and improved readability, directly addressing reviewers’ comments about figure legibility.

Overall, PatternKV offers a principled, training-free, and plug-and-play KV-cache quantization scheme centered on flattening the KV distribution via pattern-aligned residuals. It achieves near-lossless 4-bit accuracy, substantial 2-bit gains, and strong test-time scaling benefits, while adding only modest, well-quantified computational overhead and negligible memory.


**References**

[1] Zirui Liu et al. *KIVI: A Tuning-Free Asymmetric 2bit Quantization for KV Cache.* ICML 2024.

[2] Yi Su et al. *Accurate KV Cache Quantization with Outlier Tokens Tracing.* ACL 2025.

[3] Yefei He et al. *ZipCache: Accurate and Efficient KV Cache Quantization
  with Salient Token Identification.* NeurIPS 2024.

[4] Haojie Duanmu et al. *SKVQ: Sliding-window Key and Value Cache Quantization
  for Large Language Models.* COLM 2024.

---

### Author Response · Authors · 2025-12-02
**Summary of Contributions and Response to Reviewer Concerns (Part-1)**

Dear Area Chairs,

We sincerely appreciate your time and effort in handling and evaluating our submission. We thank all reviewers for their constructive feedback. Below we (i) summarize the main technical ideas and contributions of PatternKV, and (ii) clarify how we addressed the key concerns around overhead, fairness, and experimental detail.

---

## Overview and main contributions
Conceptually, PatternKV reframes KV cache quantization from “protecting outliers” to flattening the entire KV distribution:
- **Empirical analysis of KV pattern.** We conduct a detailed empirical study of KV caches, showing that (i) K caches have a remarkably stable, largely prompt-agnostic structure shaped by model-internal mappings; (ii) RoPE introduces a gradual, head-specific drift of K distributions over decoding with locally stable neighborhoods; and (iii) V caches exhibit layer-dependent semantic regularities. Beyond enabling PatternKV, these findings provide a reusable characterization of KV patterns that can inform future KV compression and architecture design, and offer an interpretable lens on how LLMs organize and store information in their KV caches.
- **Pattern-aligned residual quantization.** Based on an analysis of KV caches (Fig. 2–3 and Insights 1–3), we (i) mine a small set of per-head pattern vectors via KMeans in prefill, (ii) align each KV vector to its nearest pattern under a min–max distance, and (iii) quantize only the residual. Together with decode-time Chebyshev-center updates that track the gradual RoPE-induced drift in K distributions, this design yields substantially stronger 2-bit performance than prior KV quantization methods[1-4], while keeping 4-bit accuracy essentially on par with FP16 across representative scenarios such as long-context inference and test-time scaling.
- **Flattening-sensitive V gate.** For V cache, where semantic patterns are weaker in some mid layers, we derive a one-sided z-test based gate (Eq. 10) that only applies flattening when the contracted range $R_{\text{flat}}$ guarantees lower quantization error than raw quantization, at a chosen confidence level. Ablations show that this gate is crucial for preserving accuracy while still exploiting V patterns on the majority of tokens.
- **System design.** PatternKV is online, training-free, and plug-and-play. We implement fully GPU-parallel KMeans for prefill and two fused CUDA kernels that couple pattern lookup with QK matmul and V reconstruction, keeping the decode-time overhead of pattern selection and Chebyshev-center updates to only around 2–3% extra latency per component. Together with our pattern-aligned residual quantization, this system design reduces end-to-end latency by 30%, improves throughput by 45%, and lowers peak memory by 6% relative to the FP16 baseline.
---

## Addressing concerns on overhead and fairness

To address the reviewers’ concerns, we provide (i) a component-wise breakdown of PatternKV-specific overheads, (ii) fair, head-to-head latency/throughput and peak-memory comparisons against prior KV quantization methods [1–4], and (iii) analytical plus empirical estimates of pattern-memory cost, showing that substantial accuracy improvements over these baselines can be obtained at only modest additional overhead.

### **Component-level overhead**
We quantified the extra cost of clustering, pattern selection, and Chebyshev updates:
- **Decomposition.** Appendix J.1 breaks down the three PatternKV-specific components: prefill pattern mining (KMeans), decode-time Chebyshev-center updates, and decode-time pattern selection.
- **Measured impact (batch size 32).**
    - Pattern mining: +6.60% latency, −6.10% throughput.
    - Pattern selection: +2.59% latency, −2.47% throughput.
    - Chebyshev updates: +3.11% latency, −2.94% throughput.

The dominant overhead is the one-time KMeans clustering in prefill; decode-ime components add only ~2–3% latency each. Combined with GPU-parallel KMeans and fused CUDA kernels (Sec. 3.4), the overall per-step overhead remains modest relative to the attention pipeline.

### **Fair baseline comparison (latency / throughput)**

Several reviewers requested head-to-head comparisons with other online KV-quantization methods. Appendix J.2 reports end-to-end latency and throughput under identical hardware, model, bit-width, batch sizes, and sequence lengths:
- Compared with KIVI[1], PatternKV is somewhat slower at very large batch sizes (due to extra components) but offers significantly better accuracy.
- Compared with OTT[2], ZipCache[3], and SKVQ[4], PatternKV is competitive with OTT and consistently faster than ZipCache and SKVQ, while at the same time being more accurate across LongBench and GSM8K.
- Relative to FP16, PatternKV achieves the previously mentioned 1.4× throughput and 1.25× larger max batch on Llama-3.1-8B (Fig. 6).
We also clarified baseline configurations (Appendix F): group size, residual size, and preserved tokens are aligned as closely as possible.

---

### Meta-Review · Area_Chair_oRJe · 2026-01-06

**Summary:**

Reviewers found the methodology interesting and generally sound, with clear accuracy improvements over prior online KV-quantization methods such as KIVI, especially at low bit-widths. However, the main concern across reviews was the efficiency trade-off. While PatternKV consistently improves accuracy, it introduces non-negligible additional latency and system complexity (e.g., pattern mining, pattern selection, Chebyshev updates, and fused reconstruction kernels). Even after the rebuttal’s detailed overhead analysis and system optimizations, PatternKV remains noticeably slower than simpler baselines like KIVI, particularly at larger batch sizes, making it difficult to justify the trade-off in practical inference settings. Additional concerns included the modest net memory savings (especially under GQA), reliance on FP16 as the primary baseline, and the added engineering complexity relative to the gains. Overall, while technically interesting, reviewers remained unconvinced that the accuracy gains sufficiently outweigh the efficiency and complexity costs.

**Reviewer Concerns:**

## Concerns largely addressed in the rebuttal

- **[znsY – Weakness 1, Questions]**
  The rebuttal provides a detailed and quantitative analysis of PatternKV’s additional memory and latency overheads, including component-wise breakdowns for pattern mining (prefill), pattern selection, and Chebyshev-center updates. It also reports explicit latency, throughput, FLOPs, and peak-memory comparisons against KIVI, OTT, ZipCache, and SKVQ under matched hardware and settings.

- **[znsY – Weakness 2]**
  The authors clarify that the modest memory savings observed in the main paper are largely due to the use of GQA backbones (LLaMA-3, Qwen), and add new experiments on an MHA model (LLaMA-2-7B), where substantially larger memory and throughput gains are observed.

- **[GFhG – Reproducibility and setup clarity]**
  The rebuttal releases anonymous code, clarifies datasets and prompt templates used in analysis figures, explains residual-size choices for fair baseline comparison, adds BF16 results, and improves figure readability and experimental detail.

- **[GFhG – System efficiency questions]**
  A new system-optimization section is added, describing GPU-parallel KMeans and fused CUDA kernels, together with component-level overhead measurements and head-to-head efficiency comparisons.

## Remaining limitations

- **[znsY – Core concern on efficiency trade-off]**
  Despite the added analysis and system optimizations, PatternKV remains consistently slower than simpler baselines such as KIVI at moderate to large batch sizes. While the rebuttal acknowledges this trade-off and provides detailed measurements, it does not fundamentally change the conclusion that the accuracy gains come at a noticeable latency and throughput cost in practical inference settings.

- **[GFhG – Practical impact and trade-off]**
  Although the method is technically sound and well-motivated, the overall practical benefit remains limited in common GQA settings, where net memory savings are modest and efficiency gains over strong baselines are relatively small. The rebuttal improves clarity but does not fully resolve this concern.

- **[General]**
  The method introduces substantial system and implementation complexity (pattern mining, assignment, updates, and custom kernels) compared to prior online KV-quantization approaches. Whether this added complexity is justified by the observed accuracy improvements remains a judgment call rather than a concern conclusively resolved by the rebuttal.

**Reviewer Scores:**

- **Reviewer znsY (initial score: 4)**
  The rebuttal addressed most questions on memory overhead, latency breakdown, and fairness of comparisons, and added explicit head-to-head results with KIVI. However, the core concern regarding inferior efficiency relative to simpler baselines (especially at larger batch sizes) remains. A slight upward revision is possible, but the score would likely remain borderline (approximately 4 → 5).

- **Reviewer GFhG (initial score: 4)**
  Reproducibility, experimental clarity, and baseline fairness issues were largely addressed in the rebuttal (code release, BF16 results, clarified setups, added system analysis). Nevertheless, concerns about practical impact and the accuracy–efficiency trade-off compared to KIVI are not fully resolved. A modest increase is possible, but the score would likely remain borderline (approximately 4 → 5).

- **Reviewer 3 (initial score: 4)**
  No explicit post-rebuttal update or strong signal of a changed assessment is observed. The rebuttal does not materially alter the main trade-off concerns. The score would likely remain unchanged (4).

- **Reviewer 4 (initial score: 6)**
  This reviewer was already positive. While the rebuttal strengthens the system-level analysis, it does not introduce decisive new evidence that would clearly justify a higher score. The score would likely remain unchanged (6).

Overall, based on reviewer responses and post-rebuttal discussion, a reasonable post-discussion interpretation of the reviews is approximately **5, 5, 4, and 6**, corresponding to a **borderline reject / borderline accept** profile, with remaining concerns centered on practical efficiency relative to simpler baselines such as KIVI.

---

### Decision · Program_Chairs · 2026-01-26

Reject